# A transient mutational burst occurs during yeast colony development

Nicolas Agier [1,2], Nina Vittorelli [1,2,3], Louis Ollivier [1,2,4], Frédéric Chaux[1,2,5], Alexandre Gillet-Markowska[1,2,6], Samuel O'Donnell[1,2,7], Fanny Pouyet [1,2,4], Gilles Fischer [1,2,8 ✉] & Stéphane Delmas [1,2,8 ✉]

## Abstract

**Characterizing the contribution of mutators to mutation accumulation is essential for understanding cellular adaptation and diseases like cancer. By measuring single and double mutation rates, including point mutations, segmental duplications, and reciprocal translocations, we found that wild-type yeast colonies exhibit double mutation rates up to 17 times higher than expected from experimentally determined single mutation rates. These double mutants retained wild-type mutation rates, indicating they originated from genetically normal cells that transiently expressed a mutator phenotype. Numerical simulations suggest that transient mutator subpopulations likely consist of less than a few thousand cells, and experience high-intensity mutational bursts for less than five generations. Most double mutations accumulated sequentially across cell cycles, with simultaneous acquisition being rare and likely linked to systemic genomic instability. Additionally, we explored the genetic control of transient hypermutation and found that the excess of double mutants can be modulated by replication stress and the DNA damage tolerance pathway. Our findings suggest that transient mutators play a significant role in genomic instability and contribute to the mutational load accumulating in growing isogenic populations.**

**Keywords** Mutation Rates; Transient Mutator; Yeast; *Saccharomyces cerevisiae*; Genome Instability

**Subject Categories** Chromatin, Transcription & Genomics; Evolution & Ecology

## Introduction

Mutations arise through errors during the transmission of the genetic material over the generations, leading to permanent changes in the DNA sequence. Unfaithful repair of DNA replication errors and DNA damage induced by endogenous or exogenous stresses such as reactive oxygen species or UV, respectively, are among their main causes (Loeb, 2011). The regime and tempo of accumulation of mutations in a genome has a direct impact on the capacity of a population to adapt and to evolve. It has long been assumed that the mutation rate remains constant and is kept low but not null in a population due to a balance between various factors such as the prevalence of deleterious mutations over beneficial ones, the energetic cost of protein fidelity or the drift barrier at which even lower mutation rates are not more beneficial (Liu and Zhang, 2021; Lynch et al, 2016; Sniegowski et al, 2000).

However, over the last decade, the mutation accumulation model has departed from the classic regime of gradual accumulation of independent mutations over time towards a more complex model in which periods of mutation explosion occur between periods of gradual accumulation (Heasley et al, 2021). These bursts of mutation were characterized by the presence of mutants carrying more mutations than expected based on the mutation rate (Drake, 2007). These mutations were either clustered in the genome or dispersed over long distances along chromosomes (Drake, 2007). It was shown that, in changing environments, the cells that acquire a higher mutation rate, referred to as mutators, can be favored through natural selection because of their increased likelihood of generating adaptive mutations (Alexander et al, 2017; Giraud, 2001; Lynch et al, 2016; Matic, 2019; Sniegowski et al, 2000; Swings et al, 2017). Mutators are said to be genetic when they result from mutations in genes involved in genome maintenance. Genetic mutators are of great importance, from acquisition of antibiotics resistance in bacteria, to driver mutations in tumorigenesis (Denamur and Matic, 2006; Loeb, 2011). However, after adaptation, genetic mutators progressively lose fitness due to the increased accumulation of deleterious mutations and eventually become extinct unless they revert to lower mutation rates by obtention of rare reversion or suppressor mutations (Herr et al, 2011; Zeyl et al, 2001). By opposition to the genetic mutators that stably exert their effect over generations, another type of mutator cells called phenotypic or transient mutators was described. In bacteria and

[1]Sorbonne Université, CNRS, Computational, Quantitative and Synthetic Biology, CQSB, F-75005 Paris, France. [2]Sorbonne Université, CNRS, Inserm, Institut de Biologie Paris-Seine, IBPS, F-75005 Paris, France. [3]Collège de France, CNRS, INSERM, Centre Interdisciplinaire de Recherche en Biologie, Paris, France. [4]Université Paris-Saclay, CNRS, Laboratoire Interdisciplinaire des Sciences du Numérique, 91190 Gif-sur-Yvette, France. [5]Present address: Division for Marine and Environmental Research, Ruđer Bošković Institute, Zagreb 10000, Croatia. [6]Present address: Discngine, Paris, France. [7]Present address: Department of Plant Pathology, University of Wisconsin-Madison, Madison, Wisconsin 53706, USA. [8]These authors contributed equally: Gilles Fischer, Stéphane Delmas. ✉E-mail: gilles.fischer@sorbonne-universite.fr; stephane.delmas@sorbonne-universite.fr

yeast, it has been shown that cells with a transient mutator phenotype can originate from cell-to-cell heterogeneity in response to intrinsic or extrinsic stresses, such as starvation in quiescence or stationary phase (Pribis et al, 2019; van Dijk et al, 2015; Woo et al, 2018; Galhardo et al, 2007; Rosenberg et al, 1998; Torkelson et al, 1997; Bjedov et al, 2003; Gangloff and Arcangioli, 2017). Moreover, fluctuation in the levels of low copy number proteins involved in DNA replication and repair, caused by unequal segregation during cell division, or errors in transcription or translation can lead to the production of proteins with altered activities that can cause a temporary mutator phenotype (Drake, 2007; Ninio, 1991; Uphoff et al, 2016). In yeast, noise in the expression of genes involved in DNA replication and repair (RAD27 and RAD52) can lead to heterogeneity in recombination rate in an isogenic population (Liu et al, 2019). In *Escherichia coli*, heterogeneous, transient decreases in cellular DNA repair efficiency for a few generations have been observed within isogenic populations (Enrico Bena et al, 2024). A semi-quantitative analysis of the contribution of transient phenotypic mutators to single and double mutations in *E. coli* suggested that most double mutations would result from transient mutators (Ninio, 1991). For instance, in a clonal population of diploid *Saccharomyces cerevisiae* cells, a detailed assessment of multiple instances of loss of heterozygosity events and chromosome copy number alterations revealed the existence of subpopulations of cells capable of experiencing transient systemic genomic instability (SGI) (Heasley et al, 2020; Sampaio et al, 2020). Thus, bursts of mutations could result from the transient coexistence of heterogeneous mutation rates inside a clonal cell population (Alexander et al, 2017; Matic, 2019). However, many questions concerning the size of the mutator subpopulations, the fold increase of the mutation rate (mutational strength), the duration of the mutator episodes as well as their underlying molecular causes remain unanswered.

In this study, we uncovered the presence of transient mutator subpopulations during the development of wild-type yeast colonies that significantly contribute to the accumulation of multiple mutations in the cell. Our experimental results, supported by numerical simulations, indicate that high-intensity mutational bursts, influenced by the DNA damage tolerance pathway, occur in small subpopulations of cells. These transient mutators would influence the regime of accumulation of mutations by increasing the rates of both sequentially and simultaneously acquired double mutations.

# Results

## Rates of single and double mutations during the development of wild-type yeast colonies

We measured the rate of single and double mutations, occurring during the development of wild-type yeast colonies grown in complete medium at 30 °C, for three different types of mutations (Fig. 1A). We used two genetic systems to positively select for mutants carrying (i) a large segmental duplication of 150 kb on chromosome XV called D mutants for Duplication (Payen et al, 2008) and (ii) a reciprocal translocation between chromosomes IV and X called T mutants for Translocation (Gillet-Markowska et al,

2015). We also followed the appearance of loss of function mutations in the CAN1 gene on chromosome V (called C mutants for [Can$^R$]). These types of mutation result from different molecular pathways. The formation of the large segmental duplication relies on a Pol32-dependent replication-based mechanism (Payen et al, 2008) while the translocation results from a mitotic crossover. Both events require a homologous recombination event between a region of homology shared by two truncated heteroalleles (400 bp in *URA3* for the duplication and 410 bp in *TRP1* for the translocation, Fig. 1A). We found a complete absence of T mutant, and a 30-fold decrease of the D rate in a rad52Δ mutant background, demonstrating the importance of homologous recombination for the formation of both events (Table EV1). The C mutants mainly corresponded to the formation of point mutations resulting from unrepaired base misincorporation during DNA replication or base modification (tautomerization, deamination, alkylation), despite larger deletions or truncations could also occur. Therefore, measuring mutation rates for three different types of molecular events, allows us to monitor the overall level of genome instability that cells experienced during colony development.

We performed fluctuation assays to estimate single (D, T, and C) and double (DC and DT) mutation rates. Briefly, individual cells were deposited on solid synthetic complete medium at equal distances from each other to avoid growth interference, and incubated at 30 °C for 5 days, during which the cells divided for a total of 27 generations (Fig. 1B). Subsequently, the cells of the individual colonies were plated on selective media to count the number of single and double mutants that accumulated during the colony development. We found that the D, T, and C mutants occurred at rates of $3.3 \times 10^{-6}$, $1.8 \times 10^{-8}$, and $1.6 \times 10^{-7}$ mutations per cell per division, respectively (Fig. 2A, Table EV1). We found similar mutation rates when we grew liquid cultures for 3 days into early stationary phase before plating the cells onto selective medium (Table EV1). Moreover, these mutation rates observed in 5-day-old colonies are also similar to previously published rates for D and C mutations, also measured from liquid cultures in early stationary phase (Huang et al, 2003; Lang and Murray, 2008; Payen et al, 2008). Therefore, we found no sign of age-induced mutagenesis in our experimental set up using 5-day-old colonies, in contrast to what was described for 7-day-old *E. coli* colonies (Bjedov et al, 2003). We also showed that the three types of mutations (D, T, and C) were independent, as the presence of one type of mutation in the genome did not affect the other two mutation rates. Indeed, the T rates were not significantly different in the reference strain and in a D strain. Likewise, the D rates were identical in the reference strain and in a T or C strain. Finally, the C rates are also the same between the reference strain and a D strain (Fig. EV1A).

We tested whether some cells acquired the DC (i.e., coexistence in the same cell of the segmental duplication and the canavanine resistance) or the DT (the duplication and the reciprocal translocation) double mutations during colony development by plating the cells on media selecting for the concomitant presence of two mutations. To assess this question, a large number of individual colonies were plated on double-selective media to estimate the double mutant appearance rates (550 and 1006 colonies plated to select for DC and DT mutants, respectively, Table EV1). Overall, we obtained 16 DC and 2 DT mutants. The small number of DT double mutants obtained out of more than 1000 plated colonies reveals the detection limit of our experimental system. Based on these numbers, the observed rates of

# A

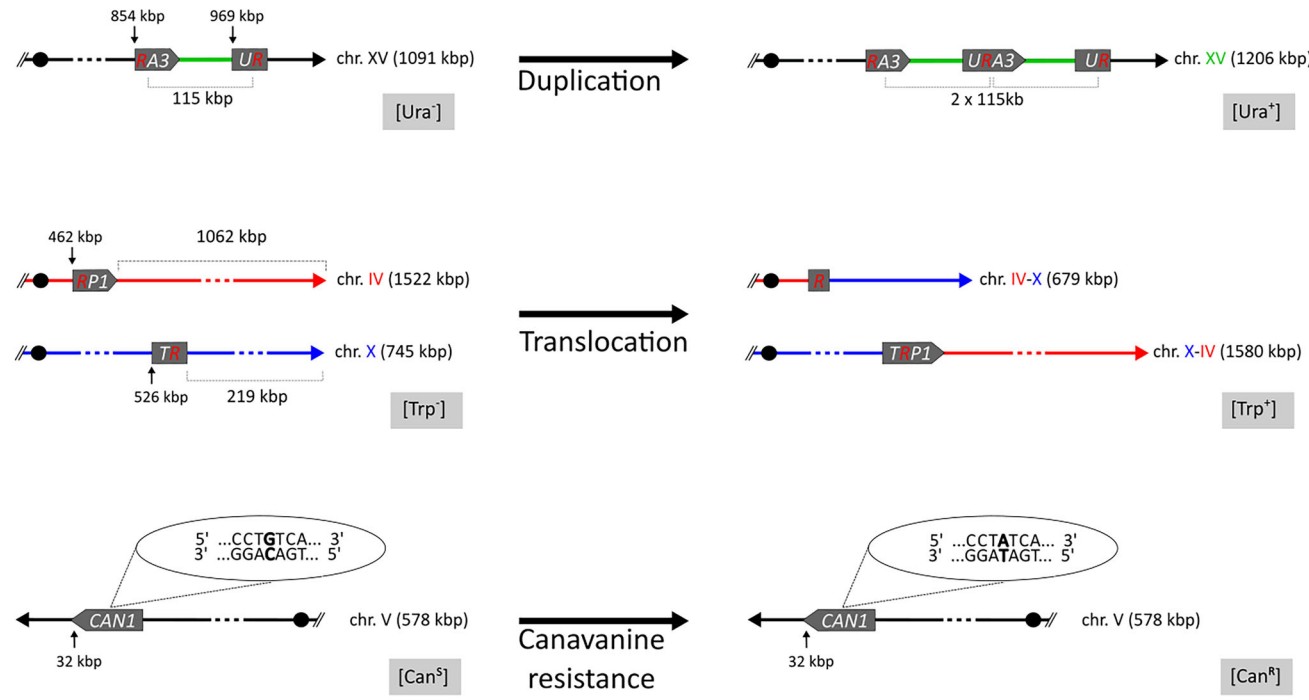

# B

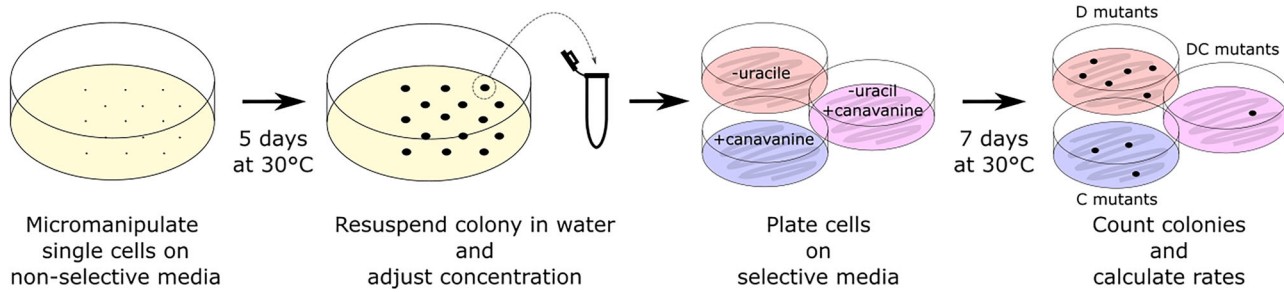

**Figure 1. Reporter systems and experimental protocol.**

(A) The Duplication system (D, top) is composed of two split alleles, "RA3" and "UR", deleted from the 5' and 3' part of the URA3 gene, respectively. These alleles are, respectively, located at coordinates 854 kb and 969 kb on chromosome XV. The two alleles share an overlapping region of 400 bp promoting, by homologous recombination, the formation of a 115 kb segmental duplication and the restoration of a functional URA3 gene at its junction. The duplication leads to an increase in size of chromosome XV to 1206 kb and a phenotypic switch from uracil auxotrophy [Ura⁻] to prototrophy [Ura⁺], allowing mutant cells to be selected on a synthetic medium lacking uracil. The Translocation system (T, middle) is composed of two split alleles, "RP1" and "TR", deleted from the 5' and 3' part of the TRP1 gene and located at coordinates 462 kb and 526 kb on chromosomes IV and X, respectively. These two alleles share an overlapping region of 410 bp promoting, by homologous recombination, the formation of a reciprocal translocation and at its junction a functional TRP1 gene. The reciprocal translocation leads to the formation of two new chromosomes IV–X and X–IV and a phenotypic switch from tryptophan auxotrophy [Trp⁻] to prototrophy [Trp⁺], allowing the selection of mutant cells on a synthetic medium lacking tryptophan. The Canavanine system (C, bottom) relies on the counter-selectable marker CAN1 gene located at coordinates 32 kb on chromosome V. Can1 is a native permease allowing the uptake of the lethal arginine homolog, canavanine. Mutation in CAN1 (symbolized here by a G to A substitution) prevent canavanine uptake and thus this loss of function leads to a phenotypic switch of the cells from canavanine sensitivity [Canˢ] to resistance [Canᴿ], allowing mutant cells to be selected on a synthetic medium lacking arginine and supplemented with canavanine. (B) Cells are deposited at equal distance from one another using a micromanipulator on a non-selective complete synthetic medium and incubated for 5 days at 30 °C. The colonies are recovered in 200 μL of sterile water, diluted if required and plated on three selective synthetic media, (i) lacking uracil, (ii) with addition of canavanine, and (iii) lacking uracil and with the addition of canavanine to select for D, C, or DC mutants, respectively. The selection for T or DT mutants on a medium lacking tryptophan or uracil and tryptophan, respectively, is not represented here. The selective plates are incubated at 30 °C before counting the mutant colonies after 7 days.

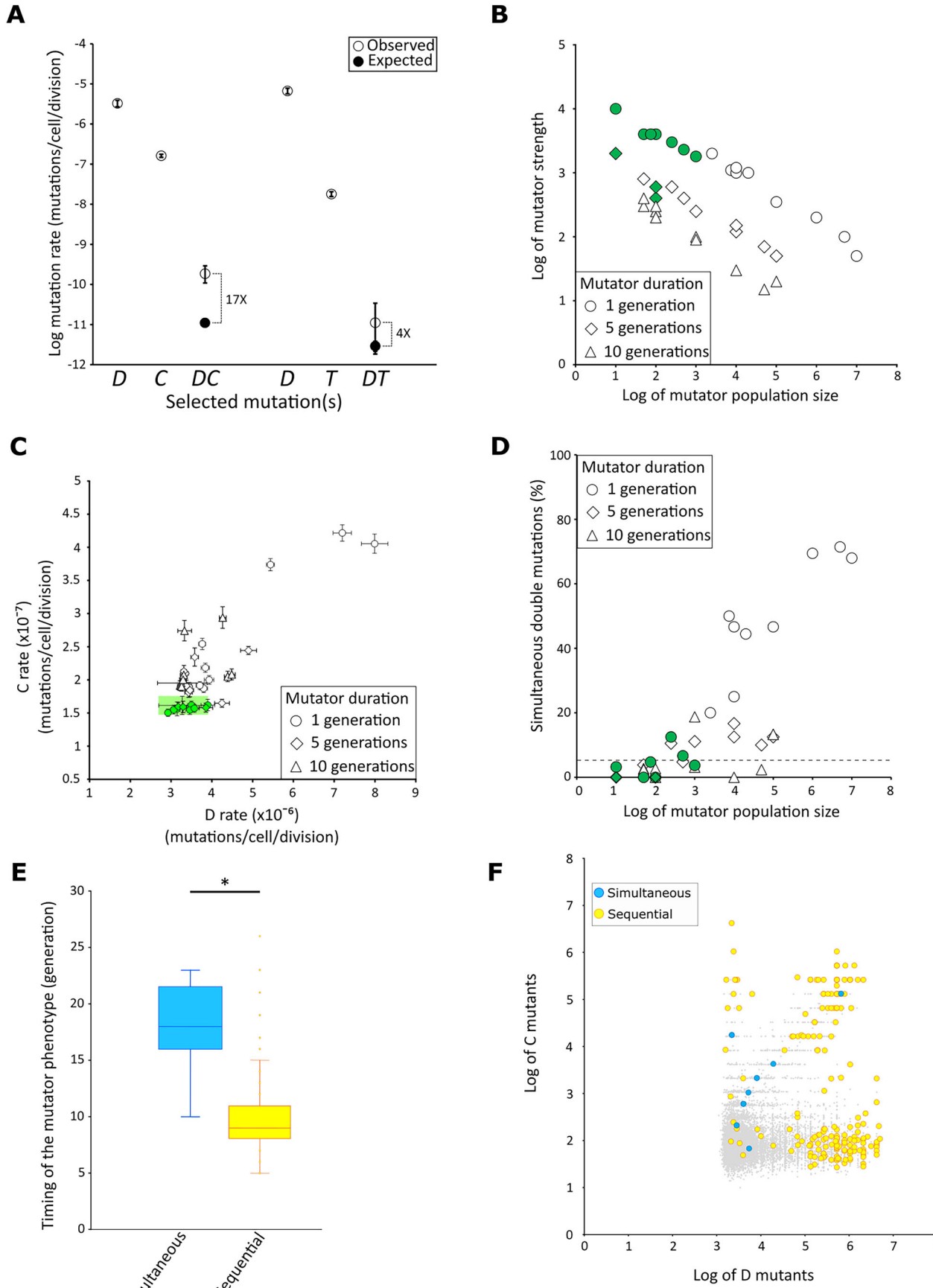

**Figure 2.  Increased double mutation rates and mutation accumulation regimes in a yeast colony.**

(A) For the reference strain (YAG142), the experimentally measured single mutation rates for the segmental duplication (D) (n = 63 independent cultures), canavanine resistance (C) (n = 104 independent cultures) and reciprocal translocation (T) (n = 229 independent cultures), and double mutation rates for duplication and canavanine resistance (DC) (n = 550 independent cultures) and duplication and translocation (DT) (n = 1006 independent cultures) are symbolized by open circles. The theoretical double mutation rates (DC) and (DT) estimated with a null model of mutation accumulation are indicated by closed circles. For each point at least 1000 realizations were performed to calculate the theoretical mutation rates. Fold changes between the observed and theoretical double mutation rates are indicated on the side. For DC (p = 1.1 × $10^{-16}$) and for DT (p = 0.04) using the likelihood ratio test. Error bars represent the 95% likelihood ratio confidence intervals, with rates. (B) Exploration of the parameter space of the refined model of mutation accumulation. On the x-axis, the mutator population size represents the number of cells that experience the transient mutator phenotype. On the y-axis, the mutator strength is expressed as a fold change increase over the observed single mutation rates. The three durations of the mutator episodes of 1, 5, or 10 generations are symbolized by open circles, diamonds, and triangles, respectively. All reported values recapitulate the experimentally observed DC double mutation rate. The points colored in green correspond to combinations of size, strength and duration of the mutator phenotype that recapitulate the experimentally observed single D and C and double DC mutation rates, as determined by the area highlighted in green in Fig. 2C. For each point at least 1000 realizations of the refined model were performed to calculate the mutation rate. (C) The D and C single mutation rates obtained from the realizations of the refined model for combinations of parameters of size, strength and duration of the mutator subpopulation presented in Fig. 2B. Open circles, diamonds and triangles correspond to the one in Fig. 2B. For each point at least 1000 realizations were performed to calculate the mutation rates. Error bars represent the 95% likelihood ratio confidence intervals. The area shaded in green corresponds to the 95% likelihood ratio confidence intervals of the experimentally measured D and C single mutation rates. The realizations that fall within this area recapitulate the experimental single mutation rates and correspond to the green points in Fig. 2B. (D) Percentage of simultaneously obtained DC mutations in the realizations of the refined model for combinations of parameters of size, strength and duration of the mutator subpopulation presented in Fig. 2B. The dotted line on the y-axis represents the percentage of simultaneous double mutants obtained from 55,000 realizations of the null model (i.e., without mutator subpopulations). The symbols colored in green are the same as in Fig. 2B, C. (E) The generation at which the mutator phenotype occurred in the realization with the refined model is represented for all realizations containing DC double mutants that were acquired either simultaneously (blue, n = 12) or sequentially (yellow, n = 195). * Student's t-test p = $10^{-15}$. The center of each box plot represents the median, the box boundaries correspond to the upper and lower quartiles, and the whiskers represent 1.5× the interquartile range. (F) Number of single D and C mutants present in a colony after a simulated growth of 27 generations, obtained from the refined model ran with parameter values that recapitulate the experimentally observed single D and C and double DC mutation rates, as determined in Fig. 2B, C (green symbols). Gray dots represent realizations having no double mutant after the 27 generations (n = 9897). Blue or yellow circles represent the realizations that comprised double mutants which were acquired either simultaneously (n = 12) or sequentially (n = 195), respectively.

DC and DT double mutant formation were $1.8 \times 10^{-10}$ and $1.1 \times 10^{-11}$, respectively (Fig. 2A). We checked that the growth rate of DT and DC double mutants was similar to that of the reference strain, ruling out the possibility that a fitness defect or advantage might have biased their appearance. Note that only one double mutant DC8 showed an increased generation time from about 120 to 180 min (Fig. EV1B). DC8 was not able to grow on a non-fermentable carbon source, and genome sequencing revealed the absence of mitochondrial DNA, indicating that it was a rho0 petite mutant (Fig. EV1C). Since the D, T, and C mutations are independent, the rate of a double mutation occurring simultaneously during the same cell division should correspond to the product of the two single mutation rates. We found that the empirically observed rates of double mutant formation were 340- and 91-fold fold higher than the product of single mutation rates for the DC and DT events, respectively. The finding that double mutants can arise dozens to hundreds of times more frequently than expected in unperturbed diploid yeast cell populations was previously reported in the context of aneuploidization and loss of heterozygosity (LOH) (Heasley et al, 2020; Sampaio et al, 2020). All these estimates rely on the assumption that all double mutations occur simultaneously during the same cell division. However, some double events could form sequentially, with one mutation occurring after the other in different cell cycles, but are not accounted for.

## Double mutations occur in large excess during the development of wild-type yeast colonies

To determine an expected double mutation rate that would take into account both the simultaneous and the sequential accumulation of the two mutations, we developed a null model of mutation accumulation that simulates the appearance of mutations during the development of the colony. This model was developed with the following assumptions: (i) cell growth is exponential, (ii) no cell death is considered, (iii) mutations appear only in daughter cells,

(iv) the two mutation rates are constant and identical for all cells in the colony, and (v) reversion mutations are ignored. This model is based on only three parameters, the two single mutation rates that we experimentally measured and the number of cell generations in the population. We performed realizations of the model to estimate the expected double mutation rate and estimate the relative contributions of simultaneous versus sequential accumulation. We found that 95% of the double mutants resulted from sequentially acquired mutations. Interestingly, the sequential formation of double LOH tracts was also observed experimentally in yeast cultures that experienced discrete episodes of systemic genomic instability (Sampaio et al, 2020). The existence of sequentially acquired double mutations implies that simply multiplying the single mutation rates significantly underestimates the actual double mutation rate. We then compared our experimentally observed double mutation rate with the expected rate that takes into account both sequentially and simultaneously acquired mutations, and found that it does not fully explain the observed double mutation rate. Thus, the observed DC double mutation rate was 17 times higher than the expected rate based on both simultaneous and sequential double mutants (p = 1.1 × $10^{-16}$, Figs. 2A and EV2). Similarly, for DT double mutants, we observed a 4 times difference between the observed and expected mutation rates (p = 0.04, Figs. 2A and EV2). Therefore, this excess of observed vs expected double mutations suggests that a subpopulation of mutator cells would generate double mutants at an increased rate during the development of yeast colonies grown in a complete medium and in the absence of exogenous stress.

## The excess double mutants result from transient phenotypic mutators

This subpopulation of mutators could originate either from stable genetic mutators or from genetically wild-type cells that express a

transient mutator phenotype. We reasoned that if the double-mutants originated from genetic mutators, they would stably inherit an elevated mutation rate, as suppressor mutation that would restore the wild-type mutation rate should be extremely rare. However, if the double mutants originated from transient mutators they would return to a wild-type mutation rate after the 27 generations that are required to isolate them from the selective medium. Thus, to determine if the double mutants came from stable or transient mutators, we measured the reciprocal transloca-tion rate in the 16 DC mutants as well as the canavanine resistance rate in the two DT double mutants. In both cases, we observed no difference in the mutation rates between the double mutants and the parental strain (Fig. EV1D). These results show that the double mutants recovered wild-type mutation rates, strongly suggesting that they originated from cells that acquired a mutator phenotype only transiently during colony development. Transient mutational bursts were first discovered and experimentally demonstrated by measuring the reversion rate of *E. coli* Lac- cells exposed to selection on lactose minimal medium (Rosenberg et al, 1998; Torkelson et al, 1997). In yeast, bursts of genomic instability were reported and also proposed to be transient based on homogeneous sequencing coverage observed in clones that acquired multiple genomic alterations over a short growth period (Heasley et al, 2021, 2020; Sampaio et al, 2020). However, to the best of our knowledge, our finding that double mutants regained a WT mutation rate represents the first direct experimental evidence of the existence of a transient mutational burst occurring during development of unperturbed yeast colonies.

We refined our stochastic model of mutation accumulation by incorporating the existence of a subpopulation of transient mutator cells during the development of the colony. This subpopulation is defined by its size (in number of cells), mutator strength (represented by the fold increase over the observed single mutation rates) and duration of the mutator phenotype (in number of generations). In this model, cells are randomly picked in the growing population to become transient mutators. We used this refined model to explore the parameters space to estimate the values that produce a double mutation rate identical to the observed DC double mutation rate that was experimentally determined. We set the duration of mutator phenotype to 1, 5, or 10 generations and varied the subpopulation size from 10 to $10^7$ cells and the mutator strength from 10- to $10^4$-fold increase. We plotted, for the three durations, all combinations of mutator sizes and strengths that recapitulated the experimentally determined DC double mutation rate (Fig. 2B). As expected, mutator strength correlates negatively with mutator size, since a smaller subpopula-tion would require a higher mutator strength to achieve the observed DC double mutation rate. For these combinations of parameters, we then recalculated the single D and C mutation rates using the simulated data and compared them to the observed single D and C mutation rates that were experimentally measured (Fig. 2C, green shaded area). We found that the introduction of the mutator subpopulation could modify the single mutation rates, as various combinations of parameters, particularly large mutator populations, led to a significant increase of the recalculated single mutation rates as compared to the experimentally observed rates (Fig. 2C). Considering that the mutation rates are kept low to avoid deleterious mutation, it has been proposed that transient mutator subpopulations should not affect the average mutation rates (Matic,

2019). In this framework, we focused on the subset of combinations of size, duration and mutator strength that replicated the experimentally measured single (D and C) and double (DC) mutation rate. For most of these combinations of parameters, the distributions of double mutants obtained in the simulations were compatible with the experimentally observed ones (Table EV2). These corresponded to parameters where the mutator durations were short (1 or 5 generations), the mutator population sizes were small (ranging from ten to thousands of cells), and the mutator strengths were high (from hundreds to $10^4$-fold increase, Fig. 2B, green symbols). Similar parameters were found for the subpopula-tions of mutators that could be responsible for the DT double mutations (Fig. EV3). Therefore, in a scenario where the mutator subpopulation does not affect the single mutation rate, the observed bursts of mutations are likely driven by small subpopulations of transient mutator cells.

## The majority of double mutations would appear sequentially during the colony development

As double mutants can either be the product of sequential or simultaneous mutations we explored the regime of acquisition of mutations in the population by performing at least 1000 realizations for each combination of parameters that recapitulates the experimentally observed DC double mutation rate. All the combinations of parameters for which simultaneously acquired mutations predominate involved large mutator population sizes resulting in single mutation rates that were higher than those measured experimentally (Fig. 2C,D). Therefore, these combina-tions were not compatible with our observations. When consider-ing only the combinations of size, duration, and mutator strength that also reproduced the observed single mutation rates, simulta-neously acquired mutations represented about 3%, and at most 12.5%, of all mutations (Fig. 2D). In conclusion, the majority of double mutants resulted from sequentially acquired mutations. The model also allows us to determine the timing of appearance of the mutations during the development of the colony. Firstly, for all realizations with a combination of parameters that are consistent with the observed single and double mutation rates and that produced double mutants after 27 generations, we determined at which generation the subpopulation acquired the mutator pheno-type. We observed that for populations with sequentially acquired mutations, the mutator phenotype appeared relatively early during the development of the colony, on average at the 9th generation (Fig. 2E). In contrast, for simultaneously acquired mutations it appeared later, on average at the 17th generation (Fig. 2E). Secondly, we plotted the number of D mutants against the number of C mutants for each simulated population (Fig. 2F). After 27 generations, the vast majority of the simulated population (98%) contained only single mutant cells (gray dots on Fig. 2F). Additionally, 91% of the colonies contained more D mutants than C mutants, in accordance with the higher mutation rate for D compared to C. Most sequentially acquired double mutants were found in colonies with a large number of D ($>10^5$) and/or C ($>10^4$) single mutants (Fig. 2F). On the contrary, most simultaneously acquired double mutants were present in colonies with a smaller number ($< 10^4$) of both D and C single mutants (Fig. 2F). These results suggest that most sequentially acquired double mutants emerge from an early mutator episode responsible for the

acquisition of a mutation (mainly *D*, due to its higher mutation rate), and subsequently the second mutation would occur in the progeny of this first mutant, independently of the initial or any mutator episode. In contrast, most simultaneously acquired mutations would result from later mutator episodes.

## Transient mutators can generate genome-wide bursts of genomic instability

To gain more insight into the two possible regimes of mutation accumulation, i.e., sequential or simultaneous, we analyzed the genomes of double mutants, reasoning that no additional mutation should be found upon the sequential accumulation of the two mutations whereas their simultaneous appearance could be associated with a general burst of genomic instability. Using Illumina short-read technology, we sequenced the genomes of 35 *DC* double mutants, including all 16 strains recovered from the fluctuation tests and 19 additional strains isolated from a mass selection scheme (Methods). A disabling mutation in the form of a frameshift, a missense or a nonsense mutation in *CAN1* was identified in every strain (Table 1), except for *DC3* (see below). The relative contribution of each type of substitution was similar to the global mutational spectrum derived from two large screens for canavanine-resistant mutants (Jiang et al, 2021; Lang and Murray, 2008). Moreover, across 14 of the 35 *DC* strains (40%), we detected a total of 27 additional and non-selected mutations, including base substitutions and indels (Table 1 and Table EV3). These 14 strains may have acquired both the selected double mutations and the non-selected additional mutations simultaneously, whereas the remaining 21 strains may have acquired the two selected mutations sequentially as they do not carry additional mutations. Interestingly, the mutational spectrum of these non-selected additional mutations deviates from the one in *CAN1*, with a higher proportion of (T:A → C:G and T:A → A:T), suggesting the existence of a mutational signature associated with transient hypermutation (Fig. EV4A). The distribution of the non-selected mutations across the 35 strains revealed that one strain (*DC3*) exhibited 11 additional mutations, three strains had two additional mutations, and 10 strains carried one extra mutation alongside the *D* and *C* mutations in their genomes (Table 1, Table EV3). These additional mutations were dispersed across different chromosomes (Table EV3). Using as parameters a genome size of $12 \times 10^6$ bp and a constant point mutation rate of $3.3 \times 10^{-10}$ substitutions per site per generation for a haploid genome (Lynch et al, 2008), we estimated that the probability to accumulate at least 1, 2, or 11 additional mutations after 27 generations was $1.0 \times 10^{-1}$, $5.3 \times 10^{-3}$, or $4.7 \times 10^{-19}$, respectively. Furthermore, the number of strains carrying those non-selected mutations significantly exceeded the number of strains that was expected under the same hypothesis (*p*-value $< 2.2 \times 10^{-16}$, Fig. EV4B). These probabilities suggest that approximately 40% of the *DC* double mutants accumulated more non-selected mutations than expected under a constant mutation rate during colony development, indicating that the colonies underwent a mutational burst.

In order to look for additional large-scale mutations, we karyotyped all *DC* (*n* = 35) and *DT* (*n* = 2) double mutants using PFGE. We found, as expected, the presence of the selected large segmental duplication on chromosome XV and reciprocal translocation between chromosomes IV and X (Fig. EV5A). Moreover, the *DC3* mutant showed an

**Table 1. List of mutations in DC mutants.**

| Strain | CAN1 mutation (in coding sequence) | Impact on Can1 protein | GCR | Number of additional mutation |
|---|---|---|---|---|
| DC1 | G → A (658) | V 220 I | D | 0 |
| DC2 | G → A (313) | G 105 S | D | 2 |
| DC3 | 30 kb deletion | Intragenic deletion | D + telomere healing | 11 |
| DC4 | −1C (1508) | T 504 fs | D | 0 |
| DC5 | C → G (311) | T 104 R | D | 0 |
| DC6 | −2T (747::748) | C 250 fs | D | 0 |
| DC7 | C → T (899) | T 300 I | D | 0 |
| DC8 | G → C (901) | E 301 Q | D | 2 |
| DC9 | C → T (356) | P 119 L | D | 0 |
| DC10 | C → A (16) | E 6 STOP | D | 0 |
| DC11 | G → A (307) | G 103 S | D | 1 |
| DC12 | G → A (1622) | W 541 STOP | D | 1 |
| DC13 | G → A (352) | G 118 S | D | 2 |
| DC14 | C → T (952) | P 318 S | D | 0 |
| DC15 | A → T (1645) | R 549 STOP | D | 0 |
| DC16 | G → T (734) C → T (1409) | C 245 F S 470 L | D | 0 |
| DC17 | C → G (1151) | T 384 R | D | 1 |
| DC21 | T → C (263) | L 88 P | D | 0 |
| DC22 | −1T (29) | I 29 fs | D | 1 |
| DC23 | T → G (682) | F 228 V | D | 1 |
| DC24 | C → T (452) | S 151 F | D | 1 |
| DC25 | G → C (508) | G 170 R | D | 1 |
| DC26 | G → A (922) | G 308 S | D | 1 |
| DC27 | C → A (996) | Y 332 STOP | D | 1 |
| DC28 | G → T (673) | E 225 STOP | D | 0 |
| DC29 | +1A (1564) | F 522 fs | D | 0 |
| DC30a | A → T (1651) | R 551 STOP | D | 0 |
| DC30b | A → C (702) | K 234 N | D | 1 |
| DC31 | C → A (996) | Y 332 STOP | D | 0 |
| DC32 | G → C (565) | G 389 R | D | 0 |
| DC33 | C → T (1267) | P 423 S | D | 0 |
| DC34 | C → A (316) | L106 I | D | 0 |
| DC35a | G → T (673) | E 225 STOP | D | 0 |
| DC35b | G → T (673) | E 225 STOP | D | 0 |
| DC35c | G → T (673) | E 225 STOP | D | 0 |

Mutations in the *CAN1* gene and non-selected additional mutations were determined by whole-genome Illumina sequencing and variant calling using GATK (Auwera and O'Connor, 2020). Gross Chromosomal Rearrangements (GCR) were visualized with PFGE and characterized by Oxford nanopore sequencing, genome assembly using Canu, NextDenovo, and SMARTdenovo (Koren et al, 2017; Liu et al, 2021; Hu et al, 2024) and SV calling using Mum&Co (O'Donnell and Fischer, 2020). *D* stands for the selected large segmental duplication of 115 kb on Chromosome XV.

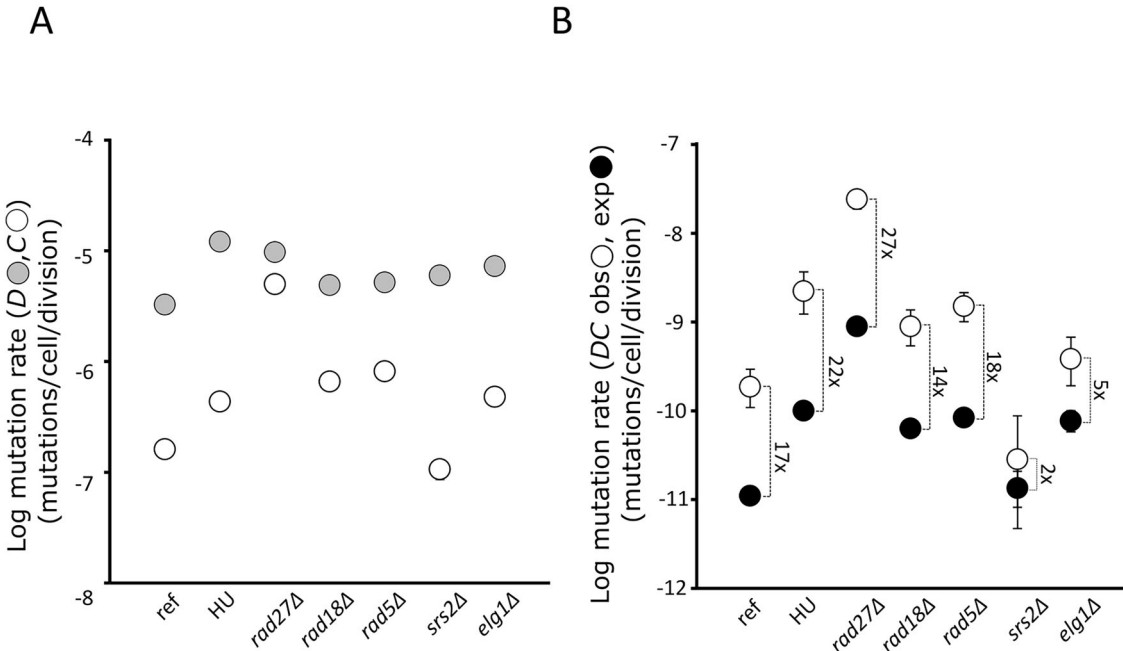

**Figure 3. Single and double mutation rates in replication stress conditions and mutant backgrounds.**

(A) The experimentally measured D (gray circle) and C (open circles) mutation rates. The reference strain (YAG142) is noted (ref). The number of independent cultures performed to measure D is as follows: ref: 63, HU: 30, rad27Δ: 30, rad18Δ: 60, rad5Δ: 58, srs2Δ: 60 and elg1Δ: 30. The number of independent cultures performed to measure C is as follows: ref: 104, HU: 30, rad27Δ: 30, rad18Δ: 60, rad5Δ: 60, srs2Δ: 60 and elg1Δ: 60. (B) The experimentally measured double DC mutation rates are symbolized by open circles. The reference strain (YAG142) is noted (ref). The number of independent cultures used to measure DC is as follows: ref: 550, HU: 236, rad27Δ: 50, rad18Δ: 149, rad5Δ: 133, srs2Δ: 493, and elg1Δ: 167. The theoretical double DC mutation rates, estimated from the null model of mutation accumulation, are indicated by closed circles. For each point at least 1000 realizations were performed to calculate the theoretical mutation rates. The fold changes between the measured and theoretical double mutation rates are indicated on the side. Error bars represent the 95% likelihood ratio confidence intervals. The genetic backgrounds of the strains and the presence of HU (100 mM) are indicated below the x-axis.

additional band on the gel (Fig. EV5A). We then fully sequenced, using Oxford Nanopore Technology, and de novo assembled the genome of 9 DC mutants, including DC3. We identified all the expected genetic markers present in the parental strain as well as the expected large segmental duplication on chromosome XV. Additionally, in the genome of DC3 we identified a terminal deletion of 30 kb on chromosome V, leading to a smaller chromosome V that corresponds to the additional band observed on the PFGE karyotype (Table 1 and Fig. EV5B). The breakpoint of the 30 kb deletion was located within the CAN1 gene where a new telomere was added using the 5'-GGTG-3'/5'CACC-3' sequence as a seed for the telomerase (Fig. EV5B). It is noteworthy that such a case of telomere healing was never observed among the 227 can1 alleles resulting from a screen for canavanine resistant mutant but is similar to the ones selected in the gross chromosomal rearrangement assay (Lang and Murray, 2008; Penna-neach et al, 2006; Putnam et al, 2004). Telomere healing at the CAN1 locus was shown to occur at a rate of about $4 \times 10^{-10}$ mutations per cell per division (Chen and Kolodner, 1999; Piazza et al, 2012). We used our null model with the rates of telomere healing and segmental duplication as parameters and estimated the probability to obtain 1 double mutant (D and terminal deletion) to be of $1.2 \times 10^{-11}$, i.e., 460 times lower than the rate at which we obtained the DC3 mutant.

Altogether, the genomic analysis of the double mutants is consistent with our model predictions which suggest that two different types of transient mutator regimes, resulting in either

sequential or simultaneous double mutants, would coexist during colony development. The strains with additional mutations likely acquired them in a single mutational episode, which may have resulted from SGI.

## Transient mutator episodes would occur in replication stress conditions

As replication stress is a major contributor to genomic instability (Kolodner et al, 2002; Toledo et al, 2017; Zeman and Cimprich, 2014), we tested whether impaired DNA replication could be responsible for the onset of the transient mutator episodes. We measured single and double mutation rates in presence of 100 mM of hydroxyurea (HU) or in a rad27Δ mutant. HU is an inhibitor of the synthesis of the deoxyribonucleotides resulting in a delayed activation of replication origins and an extended S-phase (Alvino et al, 2007), while Rad27 is involved in the processing of Okazaki fragments during DNA replication, in base excision and in mismatch repair pathways (Calil et al, 2021; Tishkoff et al, 1997). As expected for known mutator conditions, both rad27Δ and HU treatment in the reference strain significantly increased the rate of single and double mutational events compared to the reference strain (Figs. 3A,B and EV2 and Table EV1). Specifically, HU stimulates 2 to 3 times the rate for single mutation events and 12 times the DC double mutation rates (Fig. EV2 and Table EV1). Similarly, in a rad27Δ mutant, single mutation rates are increased 3

times for *D*, 30 times for *C* and 130 times for the *DC* double mutation rates (Fig. EV2 and Table EV1). Thus, double mutation rates showed clear increases under the influence of strong replication stressors. Using the neutral model, we determined the expected rate of double mutations based on the experimentally measured single mutation rates under replication stress conditions. As in the reference strain, we observed an excess of double mutants, 22-fold more in HU-treated cells and 27-fold more in the *rad27Δ* mutant than expected (Fig. 3B, Table EV1 and Fig. EV2). If transient mutator episodes were solely a consequence of replication stress, with a HU treatment or in the *rad27Δ* background, all cells in the population would be subjected to it, so the observed and expected double mutation rates should be similar. In consequence, replication stress alone does not account for the emergence of transient mutator subpopulations. In addition, the observed to expected double mutant formation rate in *rad27Δ* cells (and to a lesser extent in HU-treated cells) was higher than the 17-fold observed in the reference strain. This suggests a cumulative effect between replication stress and unknown endogenous mutagenic factors responsible for transient mutator episodes. In conclusion, the presence of mutator subpopulations in colonies grown under systemic replication stresses suggests that other factors would contribute to their emergence.

## Transient mutational bursts remain unaffected by inactivation of Rad5 or Rad18

DNA lesions causing replication fork stalling activates a surveillance pathway called the DNA Damage Tolerance (DDT) pathway, or postreplication repair pathway, that encompasses the Translesion Synthesis (TLS) and the Template Switching (TS) subpathways (Arbel et al, 2021). TLS depends on the Rad18 E3-ubiquitin ligase that monoubiquitinates PCNA and promotes the exchange of the replication polymerase by low-fidelity translesion polymerases. TS relies on the E3 Rad5 that extends polyubiquitin chains on PCNA and mediates the use of the intact sister chromatid as template to replicate across the damage. Previous studies showed that both *rad18* and *rad5* mutants exhibited increased spontaneous forward mutation rates at the *CAN1* locus as well as elevated levels of spontaneous ectopic recombination (Liefshitz et al, 1998). In accordance with these results, we found between 1.5- and 8.4-times increased *D*, *C* and *DC* formation rates in both *rad18Δ* and *rad5Δ* mutant backgrounds compared to the reference strain (Fig. 3A,B and Table EV1). Using the neutral model, we calculated the expected *DC* double mutation rate in *rad18Δ* and *rad5Δ* backgrounds. We found an excess of observed *vs* expected *DC* double mutants of 14-fold in *rad18Δ* and 18-fold in *rad5Δ* mutant backgrounds (Fig. 3B, Table EV1 and Fig. EV2). These ratios of observed to expected double mutation rates are similar to the 17-fold excess observed in the reference strain as if inactivating the TLS and TS branches of the DDT pathway had little effect, if any, on the transient hypermutation process. These results suggest that transient mutators may arise independently of TLS and TS or that functional redundancy between these pathways compensates for their inactivation. Notably, in *rad18Δ* mutants, mutations could occur via a ubiquitin-independent, Rad5-dependent error-prone repair mechanism, while in *rad5Δ* mutants, they may arise through the Rad18-dependent TLS subpathway (Arbel et al, 2021; Minesinger and Jinks-Robertson, 2005).

## Involvement of Srs2 and Elg1 in transient hypermutation

DNA lesions can also be repaired independently from TLS and TS by a Salvage Recombination (SR) subpathway. However, this mechanism is considered as potentially dangerous as it can lead to accumulation of DNA recombination intermediates and genome instability (Branzei and Szakal, 2016). The SR branch of the DDT pathway is restrained by the global inhibition of recombination at replication forks that is established through the recruitment of the Srs2 helicase to SUMOylated PCNA (Papouli et al, 2005; Pfander et al, 2005). Srs2 is a helicase that disrupts Rad51 nucleoprotein filaments, preventing unscheduled recombination events (Krejci et al, 2003; Veaute et al, 2003). Thus, Srs2 suppresses crossover and its absence leads to increased recombination frequency (Ira et al, 2003; Spell and Jinks-Robertson, 2004). We observed a twofold increase in the *D* rate in an *srs2Δ* mutant background, consistent with the requirement of homologous recombination for duplication formation (Fig. 1A). The impact of Srs2 on the point mutation rate remains largely unexplored. Here, we showed that in a *srs2Δ* mutant background the *C* rate was slightly decreased compared to the reference strain (ratio = 0.7, *p*-value = $2.7 \times 10^{-4}$, Fig. 3A, Table EV1 and Fig. EV2). This decrease is similar to the one measure in another *CAN1* assay in a *siz1Δ* mutant in which PCNA cannot be SUMOylated and therefore fails to recruit Srs2 (ratio = 0.5) (van der Kemp et al, 2009). In both cases, DNA lesions might be redirected towards the SR subpathway rather than TS or TLS (Arbel et al, 2021). We measured the *DC* double mutation rate in *srs2Δ* mutants and found that it was 6.5 times lower than the observed *DC* rate in the reference strain (Fig. 3B, Table EV1 and Fig. EV2), indicating that Srs2 activity is required to recover the double mutant. We then estimated the expected *DC* double mutation rate using the neutral model based on our experimentally determined single *D* and *C* mutation rates and compared it to the observed *DC* rate in the *srs2Δ* background. We found a 2-fold ratio of observed to expected double mutant formation rates, much smaller than the 17-fold observed in the reference strain. The low ratio of observed to expected *DC* rates suggests that the excess of double mutants generated by mutational bursts is nearly eliminated in an *srs2Δ* mutant background. This finding suggests that the formation of excess double mutants could depend on the presence of the Srs2 protein, which inhibits the SR subpathway and directs lesion bypass through TS and TLS. This supports the hypothesis that the excess double mutants arise from the functional redundancy between these two subpathways. Alternatively, the DNA lesions responsible for the formation of the excess double mutants could be toxic in an *srs2Δ* background, reducing the viability of the mutator subpopulation. Previous studies revealed that Srs2 was involved in the elimination of toxic recombination intermediates in the form of Rad51-ssDNA filaments resulting from replication fork stalling (Gangloff et al, 2000; Fabre et al, 2002; Dupaigne et al, 2008). In conclusion, the disappearance of the excess of double mutants in the srs2Δ background may result either from reduced formation or reduced viability due to an accumulation of toxic recombination intermediates.

To further test the possible implication of the SR branch in transient hypermutation, we measured single and double mutation rates in the absence of Elg1. Elg1 is a key protein at the interface of the replication and repair processes involved in the SR pathway both in *S. cerevisiae* and *Schizosaccharomyces pombe* (56–58). It is

part of the Elg1-RLC complex whose function is to unload SUMOylated forms of PCNA. Elg1 competes with Srs2 for the same binding sites on SUMOylated PCNA, thus limiting its anti-recombination activity (Arbel et al, 2021). In an *elg1* mutant background, the accumulation of SUMOylated PCNA on the chromatin generates a global replication defect, genome instability, and increased mutagenesis (Bellaoui et al, 2003; Kubota et al, 2013; Lee and Park, 2020; Shemesh et al, 2017). In line with its mutator phenotype, in our assay, *elg1Δ* mutants showed a 2- to 3-fold increase in *D*, *C*, and *DC* mutation rates compared to the reference strain (Fig. 3A,B, Table EV1 and Fig. EV2). Based on the single mutation rates, we calculated the expected *DC* mutation rate. The observed-to-expected *DC* mutation rate ratio was 5-fold, i.e., more than 3 times lower than the 17-fold observed in the reference strain. These results suggest that the formation of certain double mutants in the transient mutator subpopulation depends on the Elg1 protein.

# Discussion

This study aimed to characterize the mutation accumulation regime in isogenic cell populations. Our results revealed that the acquisition of two mutations in the genome of a cell present in a 5-day-old colony depend on the presence of transient mutator subpopulations. The presence of such subpopulations of cells reinforces the possibility that heterogeneity of mutation rate in isogenic populations could be the norm rather than the exception (Alexander et al, 2017; Matic, 2019). The existence of subpopulations of cells experiencing a transient hypermutable state was demonstrated in *E. coli* cells exposed to various exogenous stresses such as starvation on lactose (Rosenberg et al, 1998; Torkelson et al, 1997). Stress-induced mutagenesis promoted genetic instability through the activation of error prone DNA DSB repair, leading to the formation of adaptive mutations allowing cells to rapidly adapt in response to selection in their environment (Shee et al, 2011). Similarly, exposition of yeast cells to the proteotoxic drug canavanine activates the environmental stress response and induces mutagenesis through error prone DNA repair processes (Shor et al, 2013). Here, the transient mutator state that we uncovered occurred in cells within growing colonies that were not exposed to any selective environmental conditions, suggesting that this phenomenon may be more general than previously thought. Previous studies reported the increased co-occurrence of large-scale mutations, either LOH or aneuploidization, suggesting that these events underlie transient episodes of SGI in normal, clonal yeast populations (Heasley et al, 2020; Sampaio et al, 2020). Here, we specifically investigated the co-occurence of multiple types of mutational events (point mutations, large segmental duplications, and reciprocal translocations) and found that the transient mutator operates through several molecular pathways, including base substitutions and recombination-based events.

Our numerical simulations suggest that transient mutator subpopulations would be small and undergo short (1 to 5 generations) yet very intense mutational bursts, resulting in rate increases ranging from several hundred to several thousand-fold (Fig. 2). Such short periods of mutator states are consistent with what was recently observed in *E. coli* where tracking single-cell lineages revealed that temporary reductions in DNA repair efficiency can persist for 1 to 2 generations (Enrico Bena et al, 2024). This suggests that brief mutator states may be a common phenomenon across diverse organisms. In addition, constitutively high mutation rates reduce cell fitness and can lead to lineage extinction due to the accumulation of highly deleterious mutations in essential genes. Thus, mutator subpopulations may exist at the threshold of cellular viability (Funchain et al, 2000; Giraud, 2001; Herr et al, 2011). However, while some simulation parameters matching our observed data may not be compatible with long-term cell survival, particularly those with the highest mutation rates, previous studies have shown that *E. coli mutD* mutants remain viable despite a $10^4$-fold increase in mutation rate (Schaaper and Radman, 1989). Similarly, haploid yeast cells can tolerate over a thousand-fold increase in mutation rate for several generations before experiencing visible growth defects due to mutation accumulation (Herr et al, 2011). All these results indicate that the duration and strength of the mutator episodes reported here are consistent with endogenous mutagenic states that could occur within a colony.

We characterized two different regimes for the acquisition of two mutations in the genome of a cell present in a 5-day-old colony. The sequential acquisition of the two mutations is the main regime and might represent about 97% of the cases (Fig. 2D). In this regime, an early mutator episode allows the generation of a first mutation, which subsequently leads to the formation of a large subpopulation of the mutant by clonal expansion (Fig. 2E,F). The second mutation is acquired later, within the clonal mutant subpopulation but without the need for any mutator phenotype. This scenario is in agreement with the genome analysis of the *DC* double mutants that we recovered given that 21 out of the 35 clones did not carry any other mutation than the two selected ones (Table 1). We note that in human, the early acquisition of a mutation as a result of precocious mutator episodes can have huge impact in the adaptive trajectory of disease and in particular in the formation of cancer cell populations, creating mosaicism of subclones during tumorigenesis with the formation of driver and passenger mutations (Loeb, 2011; Mohiuddin et al, 2022; Seferbekova et al, 2023). The second regime corresponds to the simultaneous acquisition of the two mutations. This regime may result from an episode of SGI and may be associated with additional mutations in the genome. According to our stochastic model, this regime would occur only in a minority of cases (about 3% and up to 12.5% depending on the parameter combinations, Fig. 2D). This might represent an underestimate as 14 out of the 35 sequenced *DC* clones (40%) carried additional mutations. These included a terminal deletion associated with a new telomere addition and multiple point mutations spread on several chromosomes (Table 1, Table EV3 and Fig. EV5). These results suggest that the mutator cells endured a single SGI episode, which may result from a pathological cellular context in which mutations can accumulate in a burst during one (or a few) cell cycle(s). SGI was previously reported in diploid yeast cells growing without external stresses, by characterizing an excess of LOH events as compared to expected rates (Heasley et al, 2020; Sampaio et al, 2020). Thirty percent of the clones obtained after selecting for the two LOH events also carried extra chromosomal rearrangements, and it was proposed that 15% of the clones experiencing a LOH event in *S. cerevisiae* could do so by SGI (Sampaio et al, 2020). Interestingly, traces of SGI can potentially be found at the

population level in *S. cerevisiae*. The generation of a Reference Assembly Panel, consisting of telomere-to-telomere genome assemblies for 142 strains representatives of the entire genetic diversity of the species, revealed the presence of natural isolates with highly rearranged genomes and cases of complex aneuploidies, i.e., aneuploid chromosomes carrying additional large-scale genomic rearrangements such as translocations or large deletions (O'Donnell et al, 2023). Similarly in humans, episodes of SGI, allowing the acquisition of several mutations simultaneously, can also contribute to the transformation of human cells into a malignant state and account for the massive genome reorganization observed in tumor cells that could not be explained by a gradual accumulation (Baca et al, 2013; Campbell et al, 2020; Li et al, 2020; Pellestor, 2019; Rheinbay et al, 2020).

The presence of both early and late regimes for the acquisition of double mutations suggests that different endogenous metabolic stresses during the development of the colony could be at the origin of the transient mutator state, leading to either sequentially or simultaneously acquired mutations (Fig. 2). Indeed, it has been shown that metabolic stresses, such as carbon or amino acid starvation can induce mutagenesis (Hall, 1992; Galhardo et al, 2007; Torkelson et al, 1997; Rosenberg et al, 1998). Such stresses could originate from the temporal and spatial diversification of cellular metabolism observed in bacteria and yeast colonies (Čáp et al, 2012, 2009; Correia-Melo et al, 2023; Kamrad et al, 2023; Paczia et al, 2012; Palková et al, 2002; Palková and Váchová, 2021; Váchová et al, 2013; Saint-Ruf et al, 2014). For example, yeast colony formation is biphasic. The first phase corresponds to an exponential growth for about 24 generations (about $10^7$ cells) during which the high concentration of rDNA in the cell indicates a vigorous metabolic activity. This is followed, after 5 days, by a second phase of metabolic stress characterized by the expression of stress-induced genes, such as *SSA3* and *HSP26* (Meunier and Choder, 1999; Váchová et al, 2012). Moreover, in *S. cerevisiae* starving and stress-sensitive cells are produced during the complex reprogramming of the genetic networks and the formation of differentiated cell populations during the colony development (Čáp et al, 2012; Palková and Váchová, 2021). Furthermore, the mutator subpopulations could arise during the transition from one metabolism to another. For example, a spike of reactive oxygen species has been observed during shifts in carbon source or the transition to anoxic environments (Dirmeier et al, 2002; Koerkamp et al, 2002). If several endogenous metabolic stresses may be involved in the formation of the transient mutator subpopulation and, consequently, in the production of both sequential and simultaneous mutations, the genetic analyses of double mutant formation could help identify such stresses.

Our data show that subpopulations of mutator cells persist in a *rad27Δ* mutant, indicating that Rad27 is not necessary for the formation of double mutants. Moreover, the presence of a mutator subpopulation in *rad27Δ* mutants or in HU-treated cells, in which all cells in the population are subjected to a continuous and uniform stress, suggests that this type of stress is not involved in the formation of the transient mutator subpopulation. The precise molecular mechanisms underlying the mutational process in the transient mutator subpopulation have yet to be fully characterized, however, our findings provide new insights into the role of the DNA damage tolerance pathway. The increased formation of double mutants remains unchanged in *rad5Δ* or *rad18Δ* mutants

compared to the reference strain, but is nearly abolished in an *srs2Δ* background. These observations can be interpreted through two alternative and mutually exclusive hypotheses.

On one hand, the Srs2 protein, by dampening the checkpoint response and inhibiting recombination at damaged replication forks (Arbel et al, 2021; Fan et al, 2024; Liberi et al, 2000), may be essential for the recovery of viable double mutants from the mutator subpopulation. In its absence, toxic recombination intermediates in the form of Rad51-ssDNA filaments would accumulate (Gangloff et al, 2000; Fabre et al, 2002; Dupaigne et al, 2008), compromising the survival of mutator cells that give rise to double mutants. Thus, in an *srs2Δ* background, double mutants would be actually increasingly formed through the SR subpathway, but would not be recovered due to their reduced viability. Supporting this hypothesis, we observed fewer double mutants in an *elg1Δ* mutant compared to the reference strain (5-fold *vs.* 17-fold excess, respectively), suggesting that at least some double mutants in the reference strain rely on Elg1's ability to unload PCNA and open the SR subpathway.

On the other hand, the near-total dependency on Srs2 for double mutant recovery could indicate that these mutations arise primarily through the error-prone TS and TLS subpathways. Redundancy between these two pathways might explain why the formation of double mutants remains unchanged in *rad5Δ* or *rad18Δ* mutants relative to the reference strain. However, the reduced formation of double mutants in an *elg1Δ* background does not fully align with this second hypothesis.

In conclusion, the simulation based on our experimental data provides indication of the presence of transient mutator subpopulation during the development of yeast colonies grown in the absence of exogenous stress. This subpopulation is required for the sequential or simultaneous accumulation of two or more mutations in the genome. These findings have far-reaching implications in terms of genome evolution but further experimental validation of our computational predictions on the timing and the strength of the transient mutator episodes will be required in order to characterize the molecular triggers at the origin of these transient mutator bursts.

# Methods

**Reagents and tools table**

| Reagent/Resource | Reference or Source | Identifier or Catalog Number |
|---|---|---|
| **Experimental models** | | |
| *Saccharomyces cerevisiae* | (Baker Brachmann et al, 1998) | BY4741 |
| **Recombinant DNA** | | |
| Not applicable | | |
| **Antibodies** | | |
| Not applicable | | |
| **Oligonucleotides and other sequence-based reagents** | | |
| Not applicable | | |
| **Chemicals, Enzymes, and other reagents** | | |
| L-Canavanine | Sigma-Aldrich | C9758-5G |

| Reagent/Resource | Reference or Source | Identifier or Catalog Number |
|---|---|---|
| Hydroxyurea | Sigma-Aldrich | H8627-10G |
| Uracil | Sigma | U0750-25G |
| Leucine | Sigma | L8000-100G |
| D-glucose anhydrous | VWR | O188-5Kg |
| YPD broth | Millipore | Y1375-1Kg |
| YPD Agar | Sigma-Aldrich | A1296-500G |
| Yeast Synthetic Drop-Out media without uracil | Sigma-Aldrich | Y1501-20G |
| Yeast Synthetic Drop-Out media without tryptophan | Sigma-Aldrich | Y1876-20G |
| Yeast Synthetic Drop-Out media without uracil, leucine, tryptophan | Sigma-Aldrich | Y1771-20G |
| Yeast Synthetic Drop-Out media without leucine | Sigma-Aldrich | Y1376-20G |
| Yeast nitrogen Base without Amino Acid | Sigma-Aldrich | Y0626-250G |
| CSM—Arg-Ura | MP Biomedicals | 4523122 |
| CSM—Ura | MP Biomedicals | 4511222 |
| CSM—Arg | MP Biomedicals | 4510122 |
| SRE Kit | PacBio | 102-208-300 |
| Genomic TIP 100/G | QIAGEN | 10243 |
| Genomic DNA Buffer Set | QIAGEN | 19060 |
| QUBIT DSDNA HS ASSAYKIT, 100 | Thermo Fisher Scientific | Q32851 |
| Adapter ligation kit | Oxford Nanopore Technologies | SQK-LSK109 |
| Native barcodes | Oxford Nanopore Technologies | EXP-NBD104 |
| wash kit | Oxford Nanopore Technologies | WSH004 |
| Agencourt AMPure XP beads -5mL | Beckman | A63880 |
| Nanopore flowcell R9 | Oxford Nanopore Technologies | FLO-MIN106D |
| NEBnext companion module for Oxford Nanopore Technologies | NEB | E7180S |
| Blunt/TA ligase master mix | NEB | M0367L |
| Zymolyase 20 T | Euromedex | UZ1000 |
| RNAse A | Euromedex | 9707 |
| **Software** | | |
| R 3.6.3 / 4.1.0 / 4.3.2 | https://www.r-project.org/ | |
| R-Salvador 1.9 | (Zheng, 2017) | |
| FluCalc 2019.4.1 | (Radchenko et al, 2018) | |
| DisMut 1.0 | This study | |
| GETITE 3.4 | This study | |
| SIMPLE | This study | |
| MUM&CO 2.4.2 | (O'Donnell and Fischer, 2020) | |

| Reagent/Resource | Reference or Source | Identifier or Catalog Number |
|---|---|---|
| Snakemake 7.32.4 | (Mölder et al, 2021) | |
| BWA 0.7.18 | (Li and Durbin, 2009) | |
| GATK 4.4.0 | (Auwera and O'Connor, 2020) | |
| SLURM | (Yoo et al, 2003) | |
| CANU 2.1 | (Koren et al, 2017) | |
| GUPPY 6.1.5 | https://github.com/nanoporetech/pyguppyclient | |
| porechop 0.2.3 | https://github.com/rrwick/Porechop | |
| filtlong 0.2.0 | https://github.com/rrwick/Filtlong | |
| SMARTdenovo 1.12 | (Liu et al, 2021) | |
| NextDenovo 2.5.0 | (Hu et al, 2024) | |
| Racon 1.4.21 | (Vaser et al, 2017) | |
| Medaka 1.2.3 | https://github.com/nanoporetech/medaka | |
| Ragout 2.3 | (Kolmogorov et al, 2014) | |
| samtools 1.21 | (Danecek et al, 2021) | |
| bcftools 1.21 | (Danecek et al, 2021) | |
| Repeat Masker Tool 4.1.5 | (Smit et al, 2013) | |
| BEDtools 2.31.1 | (Quinlan and Hall, 2010) | |
| **Other** | | |
| Illumina Novaseq PE150 | Illumina | |
| MK1C Minion sequencer | Oxford Nanopore Technologies | MIN-101C |
| Qubit® 3 Fluorometer | Invitrogen | Q33216 |
| CHEF-DR III System | BioRad | 962BR1898 |
| SUNRISE spectrophotometer | Tecan Austria GmbH | 16039400 01 |

## Methods and protocols

### *Strains, growth media, and reagents*

All strains used in this study were derived from the haploid BY4741 (*MAT a, his3Δ1 leu2Δ0 met15Δ0 ura3Δ0*) (Baker Brachmann et al, 1998) and are listed in Table EV4. *RAD52*, *RAD27*, *RAD5*, *RAD18*, *SRS2*, and *ELG1* gene deletions were constructed by transformation of a PCR product containing the *KanMX4* selectable marker flanked by 50 pb of the upstream and downstream sequences of the targeted gene (Wach et al, 1994). Gene deletions were verified by growth on selective media, amplification of the insertion region by PCR and validation by Sanger sequencing. Genetic reporter systems to measure the rate of segmental duplication (*D*) described in Payen et al (2008), reciprocal translocation (*T*) described in Gillet-Markowska et al (2015) and the resistance to the canavanine (*C*) are presented in Fig. 1A. Cells were grown on empirical YPD (broth or agar) (Sigma-Aldrich) or synthetic media (glucose 20 g/L (Sigma-Aldrich), Yeast Nitrogen Based 6 g/L (Sigma-Aldrich), agar 20 g/L (Sigma-Aldrich), and complemented with specific amino acid

mixes according to the required selection. These mixes were either the "drop-out" (Sigma-Aldrich) or "CSM" (MP Biotech). When required uracil, leucine, canavanine, or hydroxyurea (all from Sigma-Aldrich) were added at 76 mg/L, 380 mg/L, 60 mg/L, or 100 mM, respectively.

### Fluctuation assays on structured and liquid medium

To measure the mutation rates in glucose-supplemented synthetic complete structured medium, strains were destocked from −80 °C on YPD agar and allowed to grow at 30 °C. After 24 h, using a micromanipulator, 15 cells per plate were individually deposited, at equal distance from each other, on a glucose supplemented synthetic complete agar medium and the plates incubated at 30 °C for 5 days (Fig. 1B). Colonies, with the exception of *petite* mutant colonies which we excluded, were individually recovered in 200 μL of sterile distilled water. In each experiment, we measured the optical density at 600 nm ($OD_{600}$) for 8 random colonies to estimate the number of cells. Cells were diluted if required, and plated on an appropriate selective synthetic medium to select for single or double mutations. Single and double mutant colonies were counted after 7 days of incubation at 30 °C. To measure the mutation rates in glucose-supplemented synthetic complete liquid medium, strains were destocked from −80 °C on YPD agar and grown at 30 °C. After 24 h a suspension of cells is prepared on complete synthetic medium, the concentration of cells determined by measuring the $OD_{600}$ and cultures inoculated with an average of 10 cells. Cultures were incubated at 30 °C with 150 rpm agitation for 3 days. After 3 days, cells were washed twice in water, then the same protocol as in structured medium is applied to select for mutant colonies.

### Measurement of mutation rates on structured and liquid medium

We systematically used 2 different methods to calculate mutation rates, the Maximum Likelihood Estimate (MLE) of *m* with the *newton.LD.plating* function using R-Salvador and the Ma-Sandri-Sarkar MSS-MLE with plating correction using FluCalc (Zheng, 2017; Radchenko et al, 2018). All rates are presented in Table EV1 and Fig. EV2. We used the likelihood ratio test to compare different mutation rate estimates (Zheng, 2017). Both methods consistently gave similar results across all estimates, only diverging with the higher mutation rates up to a maximum of 2-fold difference (Table EV1). In addition, compared to FluCalc, the *newton.LD.plating* function of R-Salvador gives mutation rates closer to the theoretical true value of mutation rates that we obtained from our stochastic model. Therefore, to be conservative, we chose to only present the rates estimated using R-Salvador in the main result section.

### Mass selection of duplication and canavanine-resistant mutants

Strains were destocked from −80 °C on YPD agar and allowed to grow at 30 °C. After 24 h, cells were suspended in water, diluted, and about 50 cells plated on a glucose supplemented synthetic complete agar medium. The plates were incubated at 30 °C for 5 days. Colonies were recovered in water, washed one time, then an equivalent of 12 colonies plated on 14.5 cm Petri dishes containing selective synthetic medium. Double mutants were recovered after 7 days of incubation at 30 °C, subcloned and stocked at −80 °C. If multiple colonies appeared on a plate, only one was selected, except in two cases, *DC30* (a and b) and *DC35* (a, b, and c), where colonies displayed noticeable differences in growth.

### Measurement of the generation time

Strains were destocked from −80 °C on YPD agar and grown at 30 °C for 24 h. Growth curves were obtained by inoculating $10^5$ cells in 200 μL of YPD broth in 96-wells plates. The plates were incubated at 28 °C without agitation in a TECAN sunrise spectrophotometer. The $OD_{600}$ was measured every 10 min. The generation time was determined using the R shiny package GETITE 3.4 available on GitHub (Data availability section).

### Pulse field gel electrophoresis (PFGE)

Whole yeast chromosome agarose plugs were prepared according to (Török et al, 1993) and sealed in a 1% Seakem GTC agarose and 0.5x TBE gel. PFGE was performed with a CHEF-DRIII (BioRad) system with the following program: 6 V/cm for 10 h with a switching time of 60 s followed by 6 V/cm for 17 h with switching time of 90 s. The included angle was 120° for the whole duration of the run. Agarose gels were stained with ethidium bromide for 20 min in TBE 0.5X.

### Short-read genome sequencing and mutation calling

Genomic DNA was extracted using a method derived from (https://doi.org/10.1038/protex.2018.076). Briefly, $2 \times 10^9$ yeast cells were harvested, washed, and resuspend in spheroblasting solution (sorbitol 1 M, $KPO_4$ 50 mM, EDTA 10 mM) with 12.5 μL of zymolyase 20 T (100 mg/ml) and 5 μL of B-mercaptoethanol. Cells were incubated for 30 min at 37 °C. Spheroblasts were recovered by centrifugation ($2500 \times g$, 3 min) then lysed for 30 min at 50 °C in 500 μL SDS-lysis buffer (TrisHcl 100 mM, EDTA 50 mM, NaCl 500 mM, PVP40 1%, SDS 2.5%) with 15 μL RNAse A (100 mM). Proteins were precipitated on ice adding 1 mL of TE buffer and 0.5 mL of KAc 5 M, followed by two successive centrifugations ($9500 \times g$, 10 min, 4 °C), keeping the supernatant. DNA was finally precipitated with 1 volume of isopropanol and washed with 70% ethanol. Resuspension was performed in water at 50 °C for 30 min. Genomic DNA was sent to BMKGENE (Munster, Germany) for sequencing using Illumina Novaseq PE150 technology.

For variant calling, we implemented a Snakemake pipeline (version 7.32.4) (Mölder et al, 2021) utilizing the following tools: BWA (0.7.18) (Li and Durbin, 2009), SAMtools (1.21) (Danecek et al, 2021), BCFtools (1.21) (Danecek et al, 2021), BEDtools (2.31.1) (Quinlan and Hall, 2010), GATK toolkit (4.4.0) (Auwera and O'Connor, 2020). The pipeline starts with fastq file mapping to a reference genome (S288C_reference_sequence_R64-4-1_20230830), followed by BAM file processing (sorting, marking duplicates, and computing coverage). Variant calling is performed with GATK's HaplotypeCaller, generating gVCF files per sample. The gVCFs are then merged and filtered into a final VCF based on per-site quality score, following GATK guidelines (QD > 20, SOR < 2, FS > 60, MQ > 50, MQRankSum > −10, BaseQRankSum > −2, −5 < ReadPosRankSum < 5). After that, repetitive regions were removed. These regions were identified by merging the output of RepeatMasker tool (version 4.1.5) (Smit et al, 2013) on the reference genome and the results from (Jubin et al, 2014). The pipeline is designed for high-throughput, parallelized processing using SLURM (Yoo et al, 2003) and was performed on the Core Cluster of the Institut Français de Bioinformatique (IFB) (ANR-11-INBS-0013). The code is available on GitHub (Data availability section).

### Long-read sequencing, de novo assembly, and genome analyses

Genomic DNA was extracted using QIAGEN genomic TIP 100/G, following the manufacturer's protocol. Size selection was done using Circulomics SRE kit (PacBio). DNA was then prepared using the ONT LSK109 kit coupled with the EXP–NBD104 barcoding kit and sequenced on a R9.1 flowcell using a MK1C Minion sequencer. Standard High Accuracy base calling was performed on the Mk1C with the 22.05.8 version of minknow, using Guppy (V6.1.5, model=dna_r9.4.1_450bps_hac.cfg). For data processing, barcodes were removed from sequences using Porechop. Then sequences were downsampled using Filtlong to obtain a coverage of 40X enriched for reads with high quality and longer length (Data availability section). Genome assembly was performed using 3 different assemblers: Canu (v2.1) (Koren et al, 2017), SMARTde-novo (Liu et al, 2021), and NextDenovo (Hu et al, 2024). Contigs were polished using 1 to 3 rounds of Racon (v1.4.21) (Vaser et al, 2017) and 2 rounds of Medaka (https://github.com/nanoporetech/medaka v1.2.3). Finally, reference Scaffolding was performed with Ragout (v2.3)(Kolmogorov et al, 2014). SVs were called using MUM&Co (v2.4.2) (O'Donnell and Fischer, 2020), on all 3 assemblies. We validated SVs found in at least 2 genome assemblies. We also checked that we detected the expected deletions corresponding to the auxotrophic markers in the BY4741 background and the insertions of resistance cassettes when applicable. For chromosomal rearrangements, manual scaffolding was performed and validated by karyotyping with PFGE.

### Modeling the theoretical distribution of the number of non-selected substitutions

We modeled the number of non-selected substitutions in a cell as a random variable following a Binomial distribution under the hypothesis of a constant mutation rate across cell divisions. We used as parameters a genome size of $12 \times 10^6$ bp, a point substitution rate of $3.3 \times 10^{-10}$ and a number of cell divisions of 27. The distribution was computed using the R library stats v. 4.3.2. The description of the mathematical framework, the code and the output are available on GitHub (Data availability section).

### A stochastic model to estimate the double mutation rate

For both the null and refined versions of the model, the number of generations and the values of the two single mutation rates are chosen by the user. For the refined model, the size of the mutator subpopulation, the fold-increase of the mutation rates and the duration of the mutator episode (in generations) are also defined by the user. The output provides the following information: (i) the number of each type of single and double mutations that occurred during the colony development, (ii) the generation at which these mutations occurred, (iii) the number of single and double mutant cells in the final colony, and (iv) the generation at which the transient mutator subpopulation starts (for the refined model only). A minimum of 1000 realizations were performed per condition. Both models are available on GitHub (Data availability section).

## Data availability

The datasets and computer code produced in this study are available in the following databases: Illumina DNA sequencing: fastq files from whole-genome sequencing of 35 *DC* strains (https://www.ebi.ac.uk/ena/browser/view/PRJEB70654). Oxford Nanopore DNA sequencing: fastq files from whole-genome sequencing of 12 strains (https://www.ebi.ac.uk/ena/browser/view/PRJEB70654). Modeling computer scripts: SIMPLE.R package to estimate the double mutation rate (https://github.com/Nico-LCQB/SIMPLE; https://doi.org/10.5281/zenodo.11183955). Modeling computer scripts: DisMut mathematical framework and code for the theoretical distribution of the number of non-selected substitutions (https://github.com/nina-vittorelli/transientBurst_additionalMutations; https://doi.org/10.5281/zenodo.13941160). Modeling computer scripts: R shiny package GETITE 3.4 to estimate growth rate (https://github.com/Nico-LCQB/GETITE; https://doi.org/10.5281/zenodo.10227629). Modeling computer scripts: Pipeline for Illumina read mapping and variant calling (https://github.com/Louis-XIV-bis/short_reads_scere; https://doi.org/10.5281/zenodo.15016541).

The source data of this paper are collected in the following database record: biostudies:S-SCDT-10_1038-S44320-025-00117-1.

## Peer review information

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

## Acknowledgements

We thank Bernard Dujon, Marina Elez, Ivan Matic, and Zhou Xu for their valuable feedback on the paper. This work was supported by the Agence Nationale de la Recherche ANR-16-CE12-0019, ANR-18-CE12-0004, and ANR-20-CE12-0020, ANR-24-CE12-0998.

## Author contributions

**Nicolas Agier**: Conceptualization; Resources; Data curation; Software; Formal analysis; Supervision; Validation; Investigation; Visualization; Methodology; Writing—original draft; Project administration; Writing—review and editing. **Nina Vittorelli**: Resources; Data curation; Software; Formal analysis; Investigation; Methodology; Writing—original draft. **Louis Ollivier**: Resources; Data curation; Software; Formal analysis; Methodology. **Frédéric Chaux**: Formal analysis. **Alexandre Gillet-Markowska**: Investigation; Methodology. **Samuel O'Donnell**: Resources; Investigation. **Fanny Pouyet**: Resources; Software; Formal analysis; Methodology. **Gilles Fischer**: Conceptualization; Supervision; Funding acquisition; Validation; Writing—original draft; Project administration; Writing—review and editing. **Stéphane Delmas**: Conceptualization; Formal analysis; Supervision; Validation; Investigation; Visualization; Writing—original draft; Project administration; Writing—review and editing.

Source data underlying figure panels in this paper may have individual authorship assigned. Where available, figure panel/source data authorship is listed in the following database record: biostudies:S-SCDT-10_1038-S44320-025-00117-1.

## Disclosure and competing interests statement

The authors declare no competing interests.

# Expanded View Figures

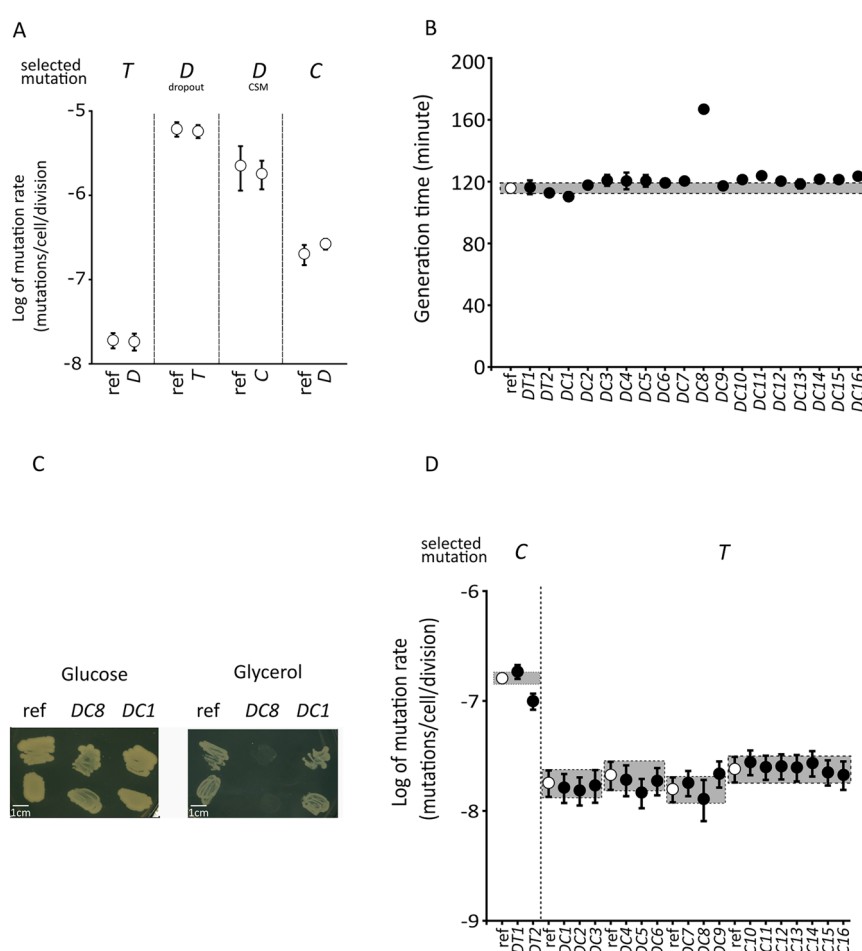

**Figure EV1. Properties of the single and double mutants.**

(**A**) Comparison of mutation rates between the YAG142 reference strain (ref) and the single mutants carrying the translocation (*T*), the segmental duplication (*D*), or the resistance to canavanine mutation (*C*). The single *T* mutation rate was measured in a *D* mutant ($n = 49$ independent cultures). The single *D* mutation rate was measured in either the *T* ($n = 49$ independent cultures) or *C* ($n = 30$ independent cultures) mutants using two synthetic media, the "dropout" (Sigma-Aldrich) or the CSM (MP Biotech). The single *C* mutation rate was measured in a *D* mutant ($n = 30$ independent cultures). Error bars represent the 95% likelihood ratio confidence intervals. (**B**) The generation time in minute of the YAG142 reference strain (ref - open circle) and the duplication and translocation (*DT*) or Duplication and Canavanine resistant (*DC*) double mutants (black circle), grown in YPD broth at 28 °C without agitation. Each point represents the average of a minimum of three replicates. Error bars represent standard deviation to the mean. (**C**) Cellular patches of the parental strain YAG142 (ref), *DC8* and *DC1* double mutants grown at 30 °C on YP-glucose (a fermentable carbon source) and YP-glycerol (a non-fermentable carbon source). (**D**) Comparison of mutation rates between the YAG142 reference strain (ref, open circles) and the double mutants carrying the duplication and translocation (*DT*) or Duplication and Canavanine resistant (*DC*, black circles). The single *C* and *T* mutation rates were measured in the 2 *DT* and 16 *DC* double mutants, respectively. The *DC* strains were tested in four different batches with each time the reference strain used as control. For each strain, a minimum of 30 independent cultures were used to measure the mutation rate. Error bars represent the 95% likelihood ratio confidence intervals.

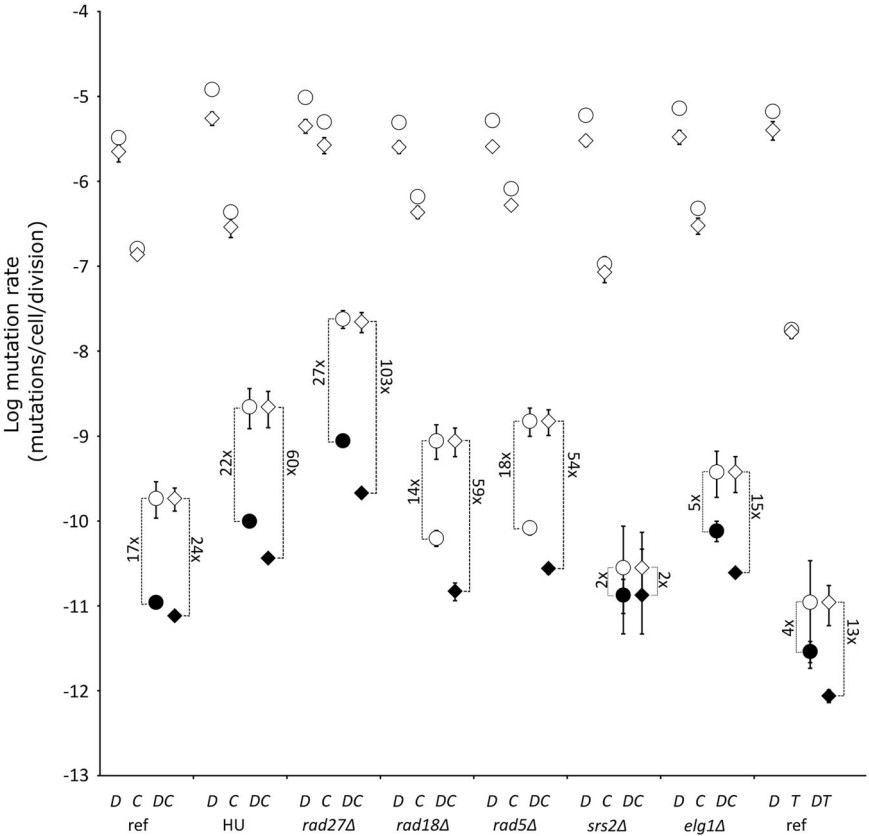

**Figure EV2. Comparison of two methods for estimating mutation rates.**

The experimentally measured single mutation rates for the segmental duplication (*D*), canavanine resistance (*C*) and reciprocal translocation (*T*), and double mutation rates for duplication and canavanine resistance (*DC*) and duplication and translocation (*DT*) are symbolized by open circles when calculated by the newton.LD.plating function of R-Salvador and open diamond when calculated by the MSS-MLE method with the web application FluCalc. The reference strain YAG142 is noted (ref). The number of independent cultures performed to measure *D* is: ref: 63, HU: 30, *rad27Δ*: 30, *rad18Δ*: 60, *rad5Δ*: 58, *srs2Δ*: 60, *elg1Δ*: 30. The number of independent cultures performed to measure *C* is: ref: 104, HU: 30, *rad27Δ*: 30, *rad18Δ*: 60, *rad5Δ*: 60, *srs2Δ*: 60, *elg1Δ*: 60. The number of independent cultures performed to measure *DC* is: ref: 550, HU: 236, *rad27Δ*: 50, *rad18Δ*: 149, *rad5Δ*: 133, *srs2Δ*: 493, *elg1Δ*: 167. For the reference strain the number of independent cultures performed to measure *T* is 229 and to measure *DT* 1006. The theoretical double mutation rates (*DC*) and (*DT*) estimated with a null model of mutation accumulation are indicated by closed circles or diamonds. For each point at least 1,000 realizations were performed to calculate the theoretical mutation rates. The fold changes between the observed and theoretical double mutation rates are indicated on the side. Error bars represent the 95% likelihood ratio confidence intervals.

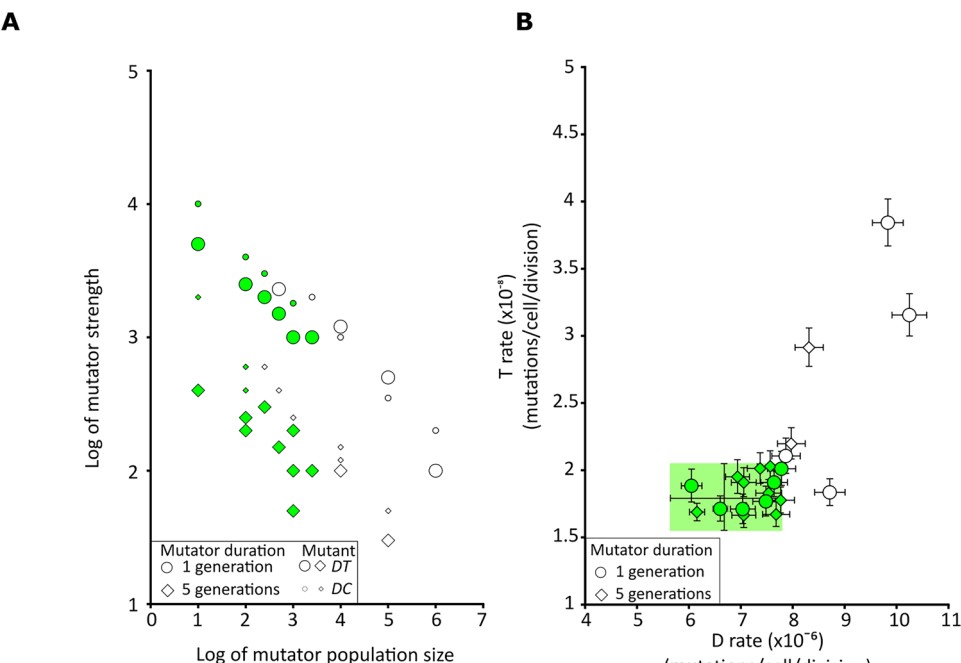

**Figure EV3. Exploration of parameter space for the duration, size, and strength of the mutator subpopulation for the formation of DT double mutants.**

(A) The mutator subpopulation size represents the number of cells that experience the transient mutator phenotype. The mutator strength is expressed as a fold change increase of the general cell mutation rate. The two mutator durations of 1 and 5 generations are symbolized by open circles and diamonds, respectively. The symbol size refers to DT (big) and DC (small) double mutants. All reported values recapitulate the experimentally observed DT and DC double mutation rates. For each point at least 1000 realizations of the refined model were performed to calculate the mutation rate. The points shaded in green correspond to combinations of size, strength, and duration of the mutator phenotype that recapitulate the experimentally observed single D, C, T, and double DC and DT mutation rates as determined by the area shaded in green in the (B) panel. (B) The symbols are as in the A panel. For each point at least 1000 realizations have been performed to calculate the mutation rates. The area highlighted in green corresponds to the 95% likelihood ratio confidence intervals of the experimentally measured D and T single mutation rates. The realizations that fall within this area recapitulate the experimental single mutation rates and therefore were colored in green in the (A) panel.

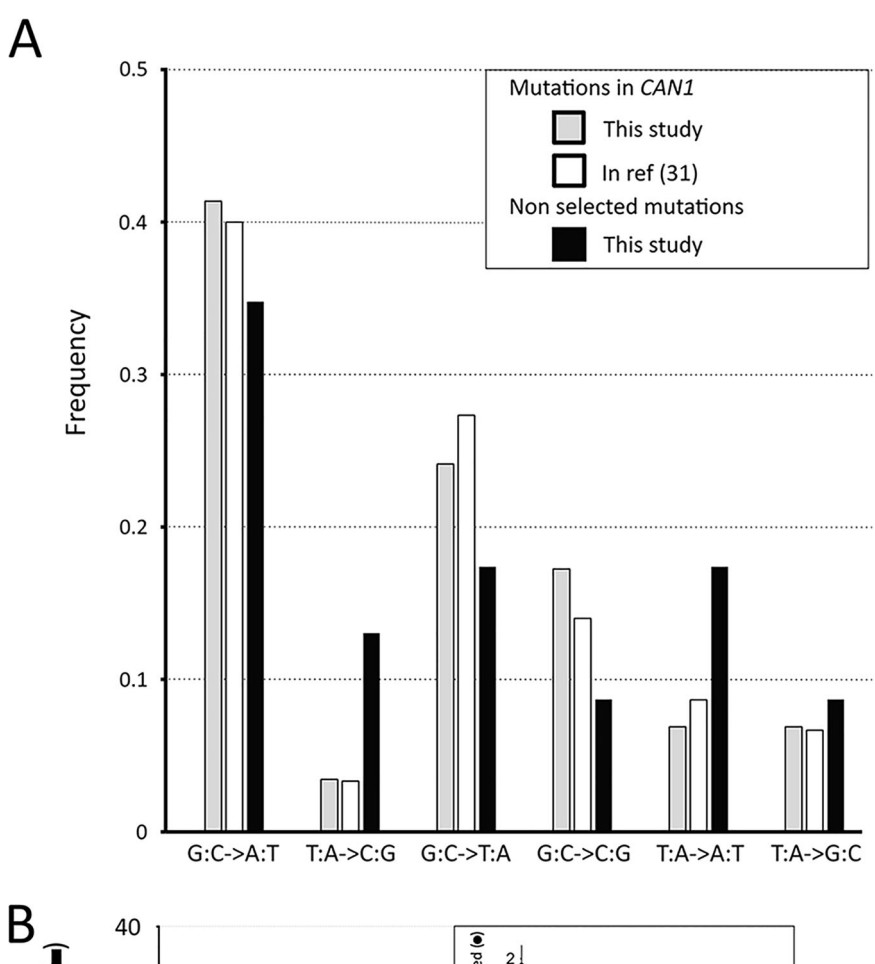

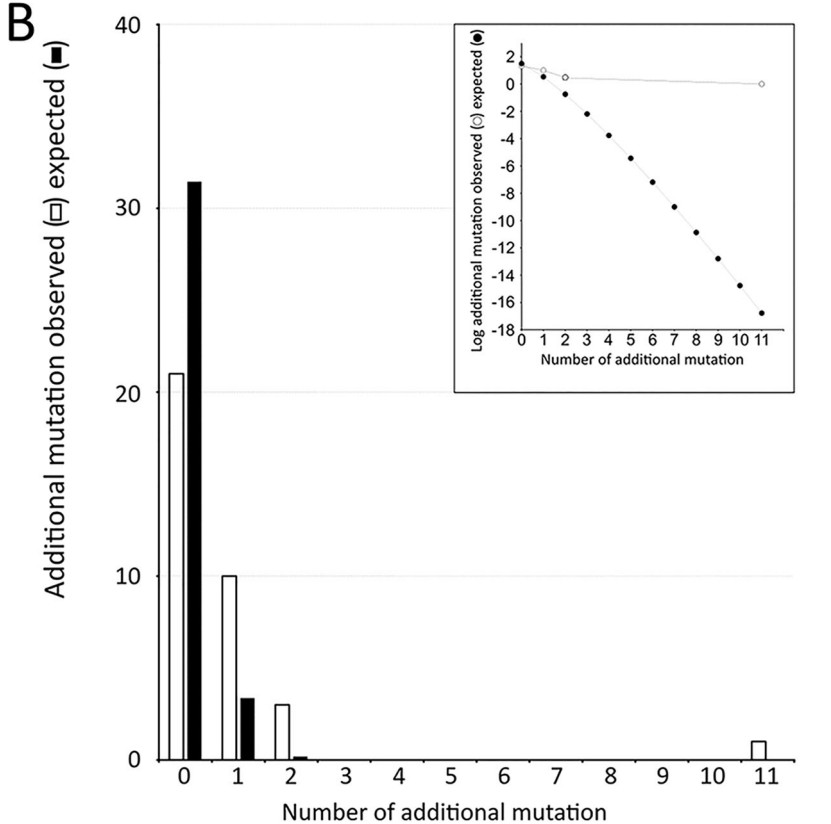

**Figure EV4. Properties of selected can1 and non-selected additional mutations.**

For both panels, the number and type of point mutations identified in this study were determined by whole-genome Illumina sequencing of the 35 *DC* mutants and variant calling using GATK (Auwera and O'Connor, 2020). (**A**) Spectrum of the selected *can1* and non-selected additional point mutations. Selected mutations in the *CAN1* gene identified in this study are symbolized by gray bars and those from reference (Lang and Murray, 2008) by open bars. The non-selected additional mutations identified in this study in the *DC* mutants are represented by black bars. (**B**) Number of observed non-selected additional mutations identified in the 35 *DC* mutants (open bar) and expected additional mutations (black bar). The numbers of expected additional mutations were determined as a random variable following a Binomial distribution, under the hypothesis of a constant mutation rate across cell divisions. The parameters used are: a genome size of $12 \times 10^6$ bp, a point substitution rate of $3.3 \times 10^{-10}$ and a number of cell divisions of 27. Inset: the same data are represented on a log scale. Comparison of the two distributions was performed by Chisquare test giving a *p*-value $< 2.2 \times 10^{-16}$. The same comparison, excluding the *DC3* mutant and its 11 additional mutations, gave a *p*-value of $1.2 \times 10^{-9}$.

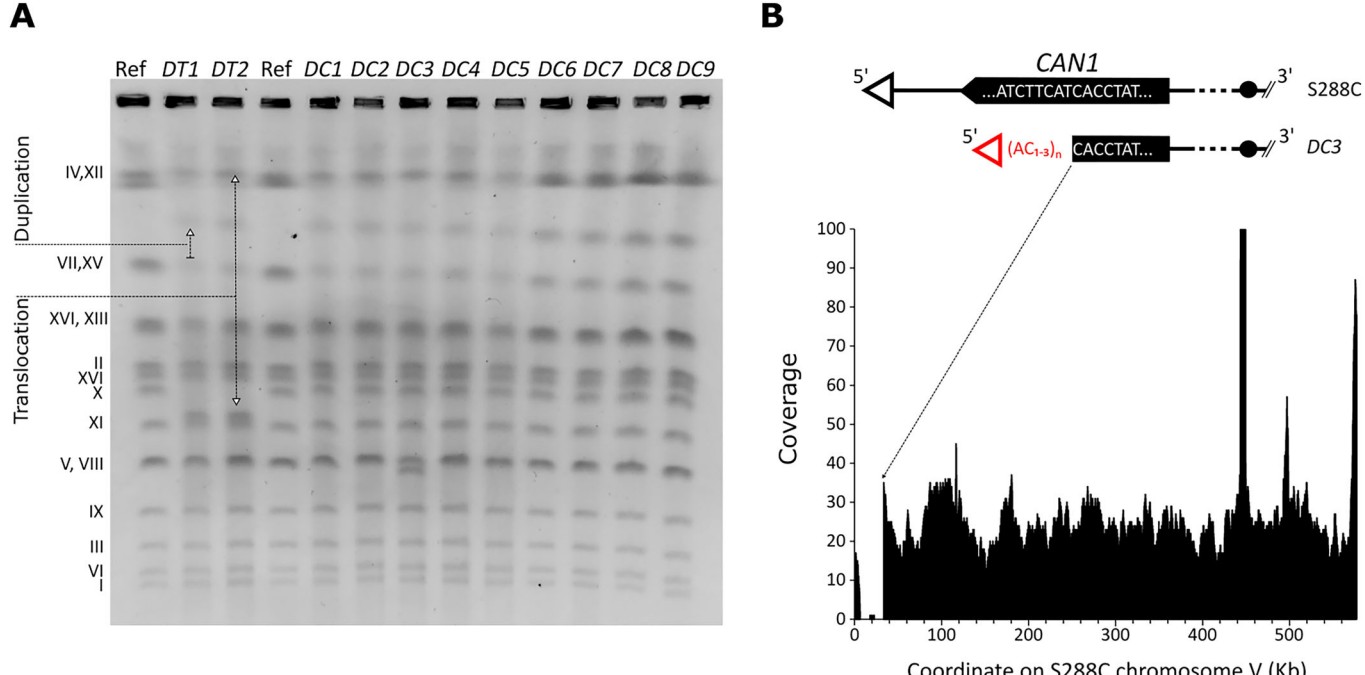

**A**

**B**

**Figure EV5. Characterization of an additional large deletion in the DC3 double mutant.**

(**A**) Pulsed-field gel electrophoresis of the parental strain YAG142 (Ref), 2 *DT* and 9 *DC* double mutants. The bands characterizing the selected reciprocal translocation and segmental duplication are indicated by arrows. The *DC3* karyotype shows a smaller band corresponding to either chromosome V or VIII. (**B**) The DNA coverage obtained from de novo genome sequencing along the chromosome V of the *DC3* strain. Upper part shows a schematic of chromosome V left arm in the reference strain S288C and in the *DC3* double mutant at the *CAN1* locus. The deletion occurred within the *CAN1* gene and a new telomere was added using a 5′-GGTG-3′/CACC-3′ seed.

