## [Peer Review File · Molecular Systems Biology]

A transient mutational burst occurs during yeast colony development

Nicolas Agier, Nina Vittorelli, Louis Ollivier, Frédéric Chaux, Alexandre Gillet-Markowska, Samuel O'Donnell, Fanny Pouyet, Gilles Fischer, and Stéphane Delmas

Corresponding author(s): Gilles Fischer (gilles.fischer@sorbonne-universite.fr), Stéphane Delmas (stephane.delmas@sorbonne-universite.fr)

Review Timeline:

Submission Date:	25th Oct 24
Editorial Decision:	20th Nov 24
Revision Received:	20th Mar 25
Accepted:	7th Apr 25

Editor: Yehu Moran

Transaction Report:

20th Nov 2024

Manuscript Number: MSB-2024-12725

Title: A transient mutational burst occurs during yeast colony development

Author: Nicolas Agier

Nina Vittorelli

Frédéric Chaux

Alexandre Gillet-Markowska

Samuel O'Donnell

Gilles Fischer

Stéphane Delmas

Dear Dr Fischer and Dr Delmas,

Thank you again for submitting your work to Molecular Systems Biology. We have now heard back from the arbitrator who agreed to evaluate your manuscript and the previous reviews. As you will see from the reports below, they find the topic of your study of potential interest. They raise, however, substantial concerns on your work, which, I am afraid to say, preclude its publication in its present form. Additionally, we stumbled by chance upon your original Referee #1 who commented as well about the manuscript. When you address the comments, please put special emphasis on Referee #2 (main Arbitrator) comments as they were not involved in the original procedure in the previous journal.

When you resubmit your manuscript, please download our CHECKLIST (<https://bit.ly/EMBOPressAuthorChecklist>) and include the completed form in your submission.

Please note that the Author Checklist will be published alongside the paper as part of the transparent process (<https://www.embopress.org/page/journal/17444292/authorguide#transparentprocess>).

If you feel you can satisfactorily deal with these points and those listed by the referees, you may wish to submit a revised version of your manuscript. Please attach a covering letter giving details of the way in which you have handled each of the points raised by the referees. A revised manuscript will be once again subject to review and you probably understand that we can give you no guarantee at this stage that the eventual outcome will be favorable.

Yours sincerely,

Yehu Moran

Academic Editor

Molecular Systems Biology

We realize that it is difficult to revise to a specific deadline. In the interest of protecting the conceptual advance provided by the work, we recommend a revision within 3 months (18th Feb 2025). Please discuss the revision progress ahead of this time with the editor if you require more time to complete the revisions. Use the link below to submit your revision:

IMPORTANT: When you send your revision, we will require the following items:

1. the manuscript text in LaTeX, RTF or MS Word format
2. a letter with a detailed description of the changes made in response to the referees. Please specify clearly the exact places in the text (pages and paragraphs) where each change has been made in response to each specific comment given
3. three to four 'bullet points' highlighting the main findings of your study
4. a short 'blurb' text summarizing in two sentences the study (max. 250 characters)
5. a 'thumbnail image' (550px width and max 400px height, Illustrator, PowerPoint or jpeg format), which can be used as 'visual title' for the synopsis section of your paper.
6. Please include an author contributions statement after the Acknowledgements section (see <https://www.embopress.org/page/journal/17444292/authorguide>)
7. Please complete the CHECKLIST available at (<https://bit.ly/EMBOPressAuthorChecklist>). Please note that the Author Checklist will be published alongside the paper as part of the transparent process (<https://www.embopress.org/page/journal/17444292/authorguide#transparentprocess>).
8. When assembling figures, please refer to our figure preparation guideline in order to ensure proper formatting and readability

in print as well as on screen:

See also figure legend guidelines: <https://www.embopress.org/page/journal/17444292/authorguide#figureformat>

9. Please note that corresponding authors are required to supply an ORCID ID for their name upon submission of a revised manuscript (EMBO Press signed a joint statement to encourage ORCID adoption).

(<https://www.embopress.org/page/journal/17444292/authorguide#editorialprocess>)

Currently, our records indicate that the ORCID for your account is 0000-0001-5732-2682.

Link Not Available

11. Include a Reagents and Tools Table as part of the Methods section, which can be downloaded from our author guidelines (<https://www.embopress.org/page/journal/17444292/authorguide#structuredmethods>)

*** PLEASE NOTE *** As part of the EMBO Press transparent editorial process initiative (see our Editorial at <https://dx.doi.org/10.1038/msb.2010.72>), Molecular Systems Biology publishes online a Review Process File with each accepted manuscripts. This file will be published in conjunction with your paper and will include the anonymous referee reports, your point-by-point response and all pertinent correspondence relating to the manuscript. If you do NOT want this File to be published, please inform the editorial office at msb@embo.org within 14 days upon receipt of the present letter.

Reviewer #1:

Thank you for inviting me to serve as "arbitrator" for this manuscript. I accepted the MSB invitation, but after reading the materials, I unfortunately will not be able to serve in this role, because I was the original Reviewer #1 for the previous submission. I am obviously biased, so I believe MSB should seek someone else to serve as arbitrator.

However, there are two comments I can offer:

1. This is the first time that I was given access to the responses to my review #1 for the original journal, and after careful consideration, I am very comfortable with the answers that the authors provided. If these responses had come back to me through the original journal, I would have recommended acceptance of the manuscript.
2. I completely agree with the statement the MSB editor wrote in the arbitration invitation email: "we believe that one of the original reviews was somewhat unfair or possibly irrelevant in some of its claims." I agree that Reviewer #2 was quite unfair in several respects, and was unjustifiably fixated on the E. coli stress-induced mutagenesis mechanism. I do not believe this is entirely appropriate to compare the E. coli work described in review #2 to the phenomenon of transient mutator states described in this manuscript.

Reviewer #2 (main arbitrator):

In this manuscript, the authors report their investigation of transient mutator populations, which have been proposed to underlie transient episodes of systemic genomic instability (SGI) occurring in normal, clonal populations of cells. The occurrence of multi-mutational bursts of genomic instability has become increasingly documented and defined over the last few years; SGI has been characterized and quantified for many types of mutation (LOH, aneuploidy, point mutations) in several systems (yeast, E. coli, human tumors). However, linking these events to a transient mutator subpopulation is difficult to address; the current models attributing SGI to transient mutators have been deduced but not demonstrated directly. Here, the authors contribute to this gap in knowledge by employing computational modeling of haploid yeast populations to predict key characteristics of these ephemeral transient mutator populations (subpopulation size, mutator strength, etc.). The main strengths and weaknesses that arose from my consideration of the manuscript are described below:

Strengths:

- 1) This work provides additional robust evidence supporting a growing body of published work demonstrating the occurrence of both a transient mutator state and bursts of mutation.

2) Whereas many (not all) previous studies of SGI examined the co-occurrence of the same class of mutations (LOH, aneuploidies, point mutations), this study investigates multi-mutational events comprised of different classes of mutation (point mutations, duplications, and translocations). I thought this was very interesting; it sheds important light on which molecular pathways could give rise to such multi-class events when perturbed. I wish that the authors had further investigated whether multi-mutational events resulting in same-class mutations occurred more/less often than multi-class events.

3) The use of computational modeling enabled the authors to describe and test key features of transient mutator populations. I find this type of modeling to be useful because it often breaks down very complex phenomena into their minimal parts. I was particularly excited about the estimates of mutator size and the duration of mutational bursts.

Weaknesses:

1) The key contribution of this work to the field was the computational modeling. In the manuscripts current form, these modeling studies suffice to define hypotheses that explain the observed rates of double mutants, but they do not demonstrate that these hypotheses are born out in real populations of cells. I appreciate that experimentally measuring some of these parameters would be challenging, but many can easily be tested in actual populations and were not. Without any form of experimental validation of the predictions made by the modeling, this report does little to advance our understanding of transient mutator populations or the causes of SGI. Most of the experiments performed in this study were necessary to establish input for the computational modeling; but alone, they are not particularly novel - the finding that double mutants arise more often than expected in unperturbed populations is already well documented.

2) The authors performed numerous fluctuation tests to calculate robust rates of single and double mutations in their system. I was surprised, then, by the low power characterization of the mutant genomes. The PFGE/nanopore analysis of the DC clones is good, but with the low number of genomes analyzed, the inferred frequencies of unselected mutations co-occurring with the selected mutations is almost certainly an underestimate - one that I believe would introduce errors into future modeling studies. PFGE and nanopore sequencing are arduous when seeking to analyze a large number of genomes; but short read illumina sequencing is not, and it can be used to detect both point mutations and copy number variations - both of which are mutations of interest in this work. I think that the contributions of this work would be vastly increased, and their conclusions would be substantially strengthened, if the authors performed a more comprehensive analysis of the mutant genomes. Doing so would almost certainly advance our understanding of whether there are mutation class biases for different co-occurring events or whether there are genomic regions that are more often affected by multi-class bursts than others.

3) The final part of the paper, in which the authors implicate ELG1 as the mechanism of transient mutator phenotypes is very intriguing, but generally lacking in substantiation. The authors neither demonstrate a mechanism by which ELG1 would induce a transient mutator phenotype, nor do they eliminate other potential (even likely) genes/pathways as other drivers of a transient mutator state. At the very least, any language implying that ELG1 is central to the appearance of transient mutators should be softened. However, I think this element of their study deserves a real mechanistic investigation that - while currently lacking - could be of great impact.

4) These authors have taken a creative new approach to investigating the very interesting topic of transient mutators. The innovativeness and novelty of this is certainly warranted. However, in this study, a number of the other conclusions drawn by the authors have previously been described by other groups. Based on the works cited in the manuscript, it is clear that the authors are aware of this, though they mostly acknowledge these previous studies in briefly in the introduction. However, I think that the transparency of the manuscript would benefit from a more directly integrated discussion of previous work through the results too. Throughout the manuscript, the reader might interpret many presented observations/conclusions as though they have just been observed for the first time, when in fact, they have been observed and characterized - albeit in slightly different systems - in the past. I think it would be much better for the readership and field in general if the authors presented their conclusions in the results section with - to highlight the differences, the similarities, etc. Not only would such a writing strategy strengthen the original findings of the authors, it would also highlight the novelty of the computational modeling approaches they use.

Editor note: I also asked the arbitrator to provide more detail about their opinion about how the authors addressed the comments from the original submission. Here is their answer:

Overall, I agreed with many of the points raised by both reviewers and the editor for the original review. A primary concern is that the authors seemed either unaware of the work - or biological context- that has been established on these topics in the past, or that they chose to downplay it to make their own findings appear more novel. More problematic in my mind was the recurrent lack of in-depth interpretation or experimental investigation. Many of the results were interpreted at face value, but not integrated into the greater scope of what is known about DNA repair and mutation; and thus none of these initial observations were mechanistically investigated with additional experiments which would have provided novel insight. I'll admit that I found the tone of the second reviewer unnecessarily hostile and struggled to see how their critiques would improve the manuscript. With that, I'll mainly comment on how the authors addressed the critiques of reviewer 1:

1) Both Reviewer 1 and I expressed that while the modeling work was very intriguing, the lack of experimental validation of the

model's predictions was a major concern. The authors claim to have addressed this by changing the last sentence of the discussion to state the importance of their modeling work and note the lack of experimental evidence. I feel that this response is not enough. Many of the other experiments in this study have been done and published with similar results several times now; the modeling work is the only advance this work contributed, yet any weight put into this modeling is substantially limited without experimental validation.

2) Reviewer 1 and I also agreed on the limited whole genome sequencing analysis that was performed on single and double mutant clones. The investigation of unselected mutations was limited to just 9 independent clones. If the authors wanted to simply corroborate that as in previous studies they also observed mutational patterns consistent with systemic genomic instability (described first by myself and colleagues in 2020), then they have achieved it. However, once again, they stopped short of performing additional analyses that could have provided new insights into the genomic regions and mutations that occur in the bursts of mutation they had selected.

3) I also agree with Reviewer 1 that the authors generally overstated the conclusions from their experiments with HU/Rad27 and ELG1 without doing any work to demonstrate mechanism. Reviewer 1 suggested that the language be softened and that the discussion of alternative interpretations be expanded - which the authors claim to have done. I feel that their modifications are inadequate. At the very least I think these sections, most crucially the last part of the results section investigating ELG1, need to be expanded to provide a stronger rationale and a more explicit discussion of the alternative interpretations of their result (of which there are many). More ideally, the authors would provide additional experimental evidence supporting their still strongly written claim that the appearance of transient mutators is dependent on ELG1. They currently provide no proposed mechanism by which ELG1 - fluctuations in its expression, posttranslational modification, something- introduces transient mutators to the population. In my perspective, this is an essential revision that needs to be performed by the authors.

Besides these major critiques, the authors did a good job responding to the reviewers issues on data accessibility.

Reviewer #1:

Thank you for inviting me to serve as "arbitrator" for this manuscript. I accepted the MSB invitation, but after reading the materials, I unfortunately will not be able to serve in this role, because I was the original Reviewer #1 for the previous submission. I am obviously biased, so I believe MSB should seek someone else to serve as arbitrator.

However, there are two comments I can offer:

1. This is the first time that I was given access to the responses to my review #1 for the original journal, and after careful consideration, I am very comfortable with the answers that the authors provided. If these responses had come back to me through the original journal, I would have recommended acceptance of the manuscript.

2. I completely agree with the statement the MSB editor wrote in the arbitration invitation email: "we believe that one of the original reviews was somewhat unfair or possibly irrelevant in some of its claims." I agree that Reviewer #2 was quite unfair in several respects, and was unjustifiably fixated on the E. coli stress-induced mutagenesis mechanism. I do not believe this it is entirely appropriate to compare the E. coli work described in review #2 to the phenomenon of transient mutator states described in this manuscript.

We sincerely thank Reviewer 1 for their continued support.

Reviewer #2 (main arbitrator):

We sincerely thank Reviewer 2 for their constructive comments, which enabled us to add highly valuable new results to our manuscript and significantly strengthen our study.

In this manuscript, the authors report their investigation of transient mutator populations, which have been proposed to underlie transient episodes of systemic genomic instability (SGI) occurring in normal, clonal populations of cells. The occurrence of multi-mutational bursts of genomic instability has become increasingly documented and defined over the last few years; SGI has been characterized and quantified for many types of mutation (LOH, aneuploidy, point mutations) in several systems (yeast, E. coli, human tumors). However, linking these events to a transient mutator subpopulation is difficult to address; the current models attributing SGI to transient mutators have been deduced but not demonstrated directly. Here, the authors contribute to this gap in knowledge by employing computational modeling of haploid yeast populations to predict key characteristics of these ephemeral transient mutator populations (subpopulation size, mutator strength, etc.). The main strengths and weaknesses that arose from my consideration of the manuscript are described below:

Strengths:

1) This work provides additional robust evidence supporting a growing body of published work demonstrating the occurrence of both a transient mutator state and bursts of mutation.

We are pleased that reviewer 2 considers that our work provides additional robust evidence to the existence of a transient mutator state.

2) Whereas many (not all) previous studies of SGI examined the co-occurrence of the same class of mutations (LOH, aneuploidies, point mutations), this study investigates multi-mutational events comprised of different classes of mutation (point mutations, duplications, and translocations). I thought this was very interesting; it sheds important light on which molecular pathways could give rise to such multi-class events when perturbed. I wish that the authors had further investigated whether multi-mutational events resulting in same-class mutations occurred more/less often than multi-class events.

In this work we purposely followed different classes of mutational events to assess whether several pathways are concerned by transient hypermutation. We agree that comparing multi-class to same-class events would also be interesting but this would require setting-up new genetic systems and performing again thousands of fluctuation tests to measure the single and double rates of same-class events, as we did for multi-class events. Thus, we believe that this is out of reach for this study.

3) The use of computational modeling enabled the authors to describe and test key features of transient mutator populations. I find this type of modeling to be useful because it often breaks down very complex phenomena into their minimal parts. I was particularly excited about the estimates of mutator size and the duration of mutational bursts.

We thank Reviewer 2 for this positive appraisal.

Weaknesses:

1) The key contribution of this work to the field was the computational modeling. In the manuscripts current form, these modeling studies suffice to define hypotheses that explain the observed rates of double mutants, but they do not demonstrate that these hypotheses are born out in real populations of cells. I appreciate that experimentally measuring some of these parameters would be challenging, but many can easily be tested in actual populations and were not. Without any form of experimental validation of the predictions made by the modeling, this report does little to advance our understanding of transient mutator populations or the causes of SGI. Most of the experiments performed in this study were

necessary to establish input for the computational modeling; but alone, they are not particularly novel - the finding that double mutants arise more often than expected in unperturbed populations is already well documented.

We agree with Reviewer 2 that the occurrence of an excess of double mutants in unperturbed populations has been previously documented, and we initially acknowledged this in both the Introduction and Discussion. However, beyond the computational modeling aspects of our work, we believe that several of our experimental findings provide novel and important controls, as well as unprecedented results. First, we demonstrate that the formation rates of *D* and *C* mutants are independent (i.e., the presence of *D* does not alter the formation rate of *C* mutants, and vice versa). This contrasts with previous studies on CCNAs, where the rate of second-chromosome loss was 2- to 12-fold higher than the euploid reference for chromosome loss (Heasley et al., Genetics 2020). This control was not included in the study focusing on double interstitial LOH events (Sampaio et al., PNAS 2020). Second, we show that the growth rates of *D* and *C* single mutants, as well as *DC* double mutants, are comparable to that of the reference strain. This finding allows us to rule out any growth-related bias in mutation rate calculations, a control absent from previous studies. Third, we demonstrate that the mutation rate in *DC* double mutants returns to wild-type levels by leveraging a third reporter system to measure the translocation rate (*T*). Although indirect, this result provides the first experimental evidence of the transient nature of the mutational burst, supporting a hypothesis previously proposed, as the reviewer mentioned. Finally, by conducting our experiments in strains deleted for master regulators of replication and repair machineries (*rad27Δ* and *elg1Δ* in the original version of the manuscript, and *rad5Δ*, *rad18Δ*, and *srs2Δ* in this revised version), our study is the first to explore the molecular pathways underlying transient hypermutation in yeast.

2) The authors performed numerous fluctuation tests to calculate robust rates of single and double mutations in their system. I was surprised, then, by the low power characterization of the mutant genomes. The PFGE/nanopore analysis of the *DC* clones is good, but with the low number of genomes analyzed, the inferred frequencies of unselected mutations co-occurring with the selected mutations is almost certainly an underestimate - one that I believe would introduce errors into future modeling studies. PFGE and nanopore sequencing are arduous when seeking to analyze a large number of genomes; but short read illumina sequencing is not, and it can be used to detect both point mutations and copy number variations - both of which are mutations of interest in this work. I think that the contributions of this work would be vastly increased, and their conclusions would be substantially strengthened, if the authors performed a more comprehensive analysis of the mutant genomes. Doing so would almost certainly advance our understanding of whether there are mutation class biases for different co-occurring events or whether there are genomic regions that are more often affected by multi-class bursts than others.

We followed the reviewer's advice by plating more than 1,500 colonies, pooling them in groups of 15, onto over 100 large petri dishes containing selective media to isolate novel *DC* double mutants. We ultimately isolated 19 novel *DC* strains, ensuring that only a single clone was retained when multiple colonies appeared on the same plate to guarantee the independence of double mutation events. The genomes of these 19 novel double mutants, along with those of the 16 initial ones described in the previous version of the manuscript, were sequenced using Illumina at high coverage to detect both point mutations and copy number variations. As now written in the revised version of the manuscript, we identified 27 additional (*i.e.* non-selected) bases substitutions and small indels in the genome of 14 out of the 35 strains (see Lines 297-324; 647-655 and 669-681, new Table EV3 and Figure EV4).

3) The final part of the paper, in which the authors implicate ELG1 as the mechanism of transient mutator phenotypes is very intriguing, but generally lacking in substantiation. The authors neither demonstrate a mechanism by which ELG1 would induce a transient mutator phenotype, nor do they eliminate other potential (even likely) genes/pathways as other drivers of a transient mutator state. At the very least, any language implying that ELG1 is central to the appearance of transient mutators should be softened. However, I think this element of their study deserves a real mechanistic investigation that - while currently lacking - could be of great impact.

We agree that the mechanistic investigation in the initial version of the manuscript was limited. To address this, we generated novel mutant strains deleted for key master genes that regulate different branches of the DNA damage tolerance pathway (*rad5Δ*, *rad18Δ*, and *srs2Δ*). We then performed fluctuation tests to measure single and double mutation rates in these new strains. These experiments provided highly informative results, demonstrating that while transient mutator episodes occur under replication stress conditions (HU and *rad27Δ*), they remain unaffected by the inactivation of Rad5 or Rad18, which control the Translesion Synthesis (TLS) and Template Switching (TS) subpathways of the DNA damage tolerance pathway. Additionally, we found that the recovery of *DC* double mutants almost entirely depends on Srs2, a helicase responsible for the global inhibition of recombination at replication forks. We have integrated these novel findings with our previous Elg1 results and now propose two alternative hypotheses that the excess double mutants would arise either through TLS/TS or Salvage Recombination. These updates are reflected in new paragraphs in the Results (Lines 381-457), Discussion (Lines 561-588), and in the revised Figure 3 and Table EV1.

4) These authors have taken a creative new approach to investigating the very interesting topic of transient mutators. The innovativeness and novelty of this is certainly warranted. However, in this study, a number of the other conclusions drawn by the authors have previously been described by other groups. Based on the works cited in the manuscript, it is clear that the authors are aware of this, though they mostly acknowledge these previous

studies in briefly in the introduction. However, I think that the transparency of the manuscript would benefit from a more directly integrated discussion of previous work through the results too. Throughout the manuscript, the reader might interpret many presented observations/conclusions as though they have just been observed for the first time, when in fact, they have been observed and characterized - albeit in slightly different systems - in the past. I think it would be much better for the readership and field in general if the authors presented their conclusions in the results section with - to highlight the differences, the similarities, etc. Not only would such a writing strategy strengthen the original findings of the authors, it would also highlight the novelty of the computational modeling approaches they use.

We thank the reviewer for this highly constructive comment. We are now discussing the results obtained in previous works throughout the Result and Discussion sections, thereby highlighting more clearly the similarities and differences with our findings (see Lines 170-178; 191-193; 218-226; 475-481; 484-487).

Editor note: I also asked the arbitrator to provide more detail about their opinion about how the authors addressed the comments from the original submission. Here is their answer:

Overall, I agreed with many of the points raised by both reviewers and the editor for the original review. A primary concern is that the authors seemed either unaware of the work - or biological context- that has been established on these topics in the past, or that they chose to downplay it to make their own findings appear more novel. More problematic in my mind was the recurrent lack of in-depth interpretation or experimental investigation. Many of the results were interpreted at face value, but not integrated into the greater scope of what is known about DNA repair and mutation; and thus none of these initial observations were mechanistically investigated with additional experiments which would have provided novel insight. I'll admit that I found the tone of the second reviewer unnecessarily hostile and struggled to see how their critiques would improve the manuscript. With that, I'll mainly comment on how the authors addressed the critiques of reviewer 1:

1) Both Reviewer 1 and I expressed that while the modeling work was very intriguing, the lack of experimental validation of the model's predictions was a major concern. The authors claim to have addressed this by changing the last sentence of the discussion to state the importance of their modeling work and note the lack of experimental evidence. I feel that this response is not enough. Many of the other experiments in this study have been done and published with similar results several times now; the modeling work is the only advance this work contributed, yet any weight put into this modeling is substantially limited without experimental validation.

As explained above, we extensively revised our manuscript to take into account the Reviewer concerns. First, we added novel experiments to characterize more thoroughly the mutant genomes by Illumina sequencing the genomes of 35 strains, 19 of them being newly isolated. Second, we constructed additional strains defective for key steps in the DNA damage tolerance pathway in order to strengthen the molecular analysis of transient hypermutation. Third, we rewrote part of the Result and the Discussion to more clearly present what was already known about transient mutations.

2) Reviewer 1 and I also agreed on the limited whole genome sequencing analysis that was performed on single and double mutant clones. The investigation of unselected mutations was limited to just 9 independent clones. If the authors wanted to simply corroborate that as in previous studies they also observed mutational patterns consistent with systemic genomic instability (described first by myself and colleagues in 2020), then they have achieved it. However, once again, they stopped short of performing additional analyses that could have provided new insights into the genomic regions and mutations that occur in the bursts of mutation they had selected.

We increased the number of sequenced genomes from 9 to 35 and added an in depth characterization of the mutational pattern associated with systemic genome instability (Table EV3 and Figure EV4).

3) I also agree with Reviewer 1 that the authors generally overstated the conclusions from their experiments with HU/Rad27 and ELG1 without doing any work to demonstrate mechanism. Reviewer 1 suggested that the language be softened and that the discussion of alternative interpretations be expanded - which the authors claim to have done. I feel that their modifications are inadequate. At the very least I think these sections, most crucially the last part of the results section investigating ELG1, need to be expanded to provide a stronger rationale and a more explicit discussion of the alternative interpretations of their result (of which there are many). More ideally, the authors would provide additional experimental evidence supporting their still strongly written claim that the appearance of transient mutators is dependent on ELG1. They currently provide no proposed mechanism by which ELG1 - fluctuations in its expression, postranslational modification, something- introduces transient mutators to the population. In my perspective, this is an essential revision that needs to be performed by the authors.

We conducted this revision by providing additional experimental evidence supporting the involvement of the DNA damage tolerance pathway in transient mutators. Additionally, we propose alternative hypotheses exploring different molecular mechanisms that could underlie the excess of double mutants.

Besides these major critiques, the authors did a good job responding to the reviewers issues on data accessibility.

7th Apr 2025

Manuscript number: MSB-2024-12725R

Title: A transient mutational burst occurs during yeast colony development

Dear Dr Fischer,

Thank you again for sending us your revised manuscript. We are now satisfied with the modifications made and I am pleased to inform you that your paper has been accepted for publication.

Yours sincerely,

Sincerely,

Yehu Moran
Academic Editor
Molecular Systems Biology

Reviewer #2:

After reviewing the revised manuscript, I have concluded that the authors have satisfactorily addressed the previous critiques. The newest version is vastly improved in how it contextualizes the new results. It also contains new data - the expanded sequencing based analyses, specifically - that contribute novel insight into the patterns of transient mutator events that occur in populations.
